

# Probing into the aging dynamics of biomass burning aerosol by using satellite measurements of aerosol optical depth and carbon monoxide

Igor B. Konovalov[1], Matthias Beekmann[2], Evgeny V. Berezin[1], Paola Formenti[2], Meinrat O. Andreae[3,4]

[1]Institute of Applied Physics, Russian Academy of Sciences, Nizhniy Novgorod, Russia
[2]LISA/IPSL, Laboratoire Interuniversitaire des Systèmes Atmosphériques, UMR CNRS 7583, Université Paris Est Créteil (UPEC) et Université Paris Diderot (UPD), France
[3]Biogeochemistry Department, Max Planck Institute for Chemistry, Mainz, Germany
[4]Scripps Institution of Oceanography, University of California San Diego, La Jolla, CA 92093, USA

*Correspondence to*: Igor B. Konovalov (konov@appl.sci-nnov.ru)

**Abstract.** Carbonaceous aerosol released into the atmosphere from open biomass burning (BB) is known to undergo considerable chemical and physical transformations (aging). However, there exists substantial controversy about the nature and observable effects of these transformations. A shortage of consistent observational evidence on BB aerosol aging processes in different environmental conditions and at various temporal scales hinders developing their adequate representations in chemistry transport models (CTMs). In this study, we obtain insights into the BB aerosol dynamics by
using available satellite measurements of aerosol optical depth (AOD) and carbon monoxide (CO). The basic concept of our method is to consider AOD as a function of the BB aerosol "photochemical age" (that is, the time period characterizing the exposure of BB aerosol emissions to atmospheric oxidation reactions) predicted by means of model tracers. We evaluate the AOD enhancement ratio (ER) defined as the ratio of optical depth of actual BB aerosol with respect to that of a modeled aerosol tracer that is assumed to originate from the same fires as the real BB aerosol but is not affected by any aging
processes. To limit possible effects of model transport errors, the AOD measurements are normalized to CO column amounts that are also retrieved from satellite measurements. The method is applied to the analysis of the meso- and synoptic-scale evolution of aerosol in smoke plumes from major wildfires that occurred in Siberia in summer 2012. AOD and CO retrievals from, respectively, MODIS and IASI measurements are used in combination with simulations performed with the CHIMERE CTM. The analysis indicates that aging processes strongly affected the evolution of BB aerosol in the situation
considered, especially in dense plumes (with $PM_{2.5}$ concentration exceeding 100 µg m$^{-3}$). For such plumes, the ER is found to increase almost twofold on the scale of ~ 10 hours of the daytime evolution of aerosol (after a few first hours of the evolution that are not resolved in our analysis). The robustness of this finding is corroborated by sensitivity tests and Monte Carlo experiments. Furthermore, a simulation using the volatility basis set framework suggests that a large part of the increase in the ER can be explained by atmospheric processing of semi-volatile organic compounds. Our results are
consistent with findings of a number of earlier studies reporting considerable underestimation of AOD by CTMs in which BB aerosol aging processes have either been disregarded or simulated in a highly simplified way. In general, this study demonstrates the feasibility of using satellite measurements of AOD in biomass burning plumes in combination with aerosol





tracer simulations for the investigation of BB aerosol evolution and validation of BB aerosol aging schemes in atmospheric models.

# 1 Introduction

Aerosol from open biomass burning (BB) is an important agent of the climate and weather systems, which affects the optical characteristics and thermal balance of the atmosphere both directly, by absorbing and scattering incoming solar radiation, and indirectly, by modifying cloud properties (e.g., Haywood and Boucher, 2000; Andreae and Merlet; 2001; Pierce et al., 2007, Bond et al., 2013; Andreae and Ramanathan, 2013; Alonso-Blanco et al., 2014). It is also an important factor in the context of air pollution, as smoke plumes from wildfires and other types of BB have been reported to drastically degrade air quality in many regions worldwide (e.g., Bertschi and Jaffe, 2005; Konovalov et al., 2011; Strand et al., 2012; Engling et al., 2014).

The two major carbonaceous fractions of BB aerosol that are conventionally considered to characterize its physical and chemical properties are black (or elemental) carbon and organic aerosol (or organic carbon). Black carbon (BC) constituting typically 2-10% of the mass concentration of biomass burning aerosol (see, e.g., Reid et al., 2005a and references therein) is commonly implied to represent the major light-absorbing component of BB aerosol and is typically composed of graphite-like structures that predominantly consist of carbon atoms (Andreae and Gelencsér, 2006). Organic aerosol (OA) commonly makes up most of BB aerosol particle mass and consists of thousands of individual organic compounds covering a wide range of physical and chemical properties. Typically, OA features much lower mass absorption efficiency than BC and contributes to light scattering in the atmosphere (see, e.g., Keil and Haywood, 2003); however, OA in BB plumes is also known to have a low-volatility light-absorbing fraction called "brown carbon" (BrC) that (similar to BC) has been shown to be an important factor in aerosol radiative forcing (Andreae and Ramanathan, 2013; Saleh et al., 2014). Apart from the carbonaceous fractions, BB aerosols contain a typically minor (of order of 10%) mass fraction of inorganic species, such as potassium, sulfate, chloride, etc. (Reid et al., 2005a).

There is abundant evidence that BB aerosol undergoes considerable chemical and physical transformations that are commonly referred to as aerosol "aging". For example, substantial enhancements of the oxygenated organic fraction of wood burning aerosol were observed by Grieshop et al. (2009), Hennigan et al. (2011) and Heringa et al. (2011) in smog chamber experiments after several hours of photochemical processing of biomass burning emissions. Consistently with the laboratory experiments, mass-spectroscopic measurements acquired during several aircraft campaigns in different regions of North America (DeCarlo et al., 2008; Cubison et al., 2011; Jolleys et al., 2015; May et al., 2015) revealed that the BB OA is usually more oxygenated downwind of fires than in fresh smoke plumes. Aging of BB aerosol has also been shown to be associated with increasing single scattering albedo (Reid et al., 1998; Abel et al., 2003), decreasing real and imaginary parts of the complex refractive index (Adler et al., 2011), and evolution of the number size distribution (Dubovik et al., 2002; Formenti et al., 2003; Capes et al., 2008).





There is, however, substantial controversy regarding the effects of aging on the mass concentration of BB aerosol. Such effects are commonly evaluated by means of the normalized excess mixing ratio (NEMR) or enhancement ratio (ER) both of which, in the given context, represent the ratio of mass concentration of BB aerosol or its component, such as OA, to that of the BB fraction of a long-lived tracer (such as BC, CO or $CO_2$). On the one hand, smog chamber experiments indicated that

the ER of OA with respect to BC typically increases after a few hours of aging, although there has been significant variability among results of individual experiments, some of which showed a strong increase of this ratio (up to a factor of 4), while some others resulted in a net loss of OA (Grieshop et al., 2009; Hennigan et al., 2011; Heringa et al. 2011). A strong increase (by more than a factor of 2) of the ER for $PM_{2.5}$ with respect to CO during first two hours of the atmospheric evolution of smoke plumes from deforestation and crop residue fires on the Yucatan peninsula was detected in aircraft

measurements (Yokelson et al., 2009). DeCarlo et al. (2010) estimated (also using aircraft measurements) that aging of BB plumes around Mexico City results in adding OA mass equivalent to about 32–42% of the primary OA emissions over several hours to a day. A steep growth of the ER for the mass concentration of submicron aerosol ($PM_1$) originating from fires in African savannas during the first few hours of atmospheric transport was reported by Vakkari et al. (2014). Reid et al. (1998) estimated that aerosols from fires in the Amazon region grew in mass by about from 20 to 40% during much

longer evolution (over one to four days). Recently, Konovalov et al. (2015) found observational evidence for a substantial increase (by a factor of 2) of the ER for $PM_{10}$ in smoke plumes that have been transported from a source region in the European part of Russia to Finland during one to two days.

On the other hand, there is evidence across several field studies that aging of BB plumes is not necessarily connected with a net increase in OA mass. In particular, Capes et al. (2008) found the relationship between BB OA mass concentration and

CO concentration measured by aircraft over Western Africa to be almost unaffected by BB plume ages. Similarly, analyzing aircraft measurements of BB plumes from wildfires in Canada, Sakamoto et al. (2015) found no detectable evidence of production or loss of OA mass within the aged BB plumes (with an age of 1-2 days) from wildfires in Canada. Akagi et al. (2012) reported a net decrease of OA by about 20% over 4 hours of aging of BB emissions from chaparral fires in California, although an initial sharp decrease of the ER for OA was followed by its gradual increase after ~ 2 hours of the evolution.

Jolleys et al. (2012) calculated ERs for OA from ambient measurements recorded during four field campaigns in Australia, western Africa and North America and found that their values in fresh plumes (identified by means of an indirect criterion based on the magnitude of the OA concentration) had been consistently larger compared to those in aged plumes. More recently, Jolleys et al. (2015) reported a similar finding from aircraft measurements of BB plumes from wildfires in Canada, although they pointed out that the observed difference between ERs for fresh and aged plumes could be influenced by a

change in dominant combustion conditions throughout the campaign.

Since it is evident that transformations of BB aerosol properties associated with atmospheric processing can be very significant, adequate representation of BB aerosol ageing in chemistry transport and climate models is an essential prerequisite for reliability of their results. Meanwhile, a number of studies involving comparisons of observations of BB





aerosol with simulations revealed major systematic discrepancies that may be indicative of serious shortcomings of model schemes commonly used to simulate BB aerosol evolution. In particular, atmospheric models have been found to underestimate BB aerosol optical depth (AOD) observed in different regions of the world (e.g., Matichuk et al., 2008; Johnson et al., 2008; Colarco et al., 2010; Kaiser et al., 2012; Tosca et al., 2013; Petrenko et al., 2012; Konovalov et al.,

2014, Reddington et al., 2016). The underestimations are commonly corrected by manual enhancements in BB aerosol emissions. For example, in order to match simulations and satellite observations of AOD, Tosca et al. (2013) doubled biomass burning carbonaceous aerosol emissions based on the GFEDv3 inventory (van der Werf, 2010) by a factor of 2 globally and applied additional regional scaling factors ranging from 1.45 (in North America) to 2.4 (in South America). While introducing the GFAS v1.0 fire emission inventory, Kaiser et al. (2012) recommended increasing the BB emissions of

particulate matter globally by a factor of 3.4 in order to ensure a reasonable agreement between observed and modeled values of AOD over major BB regions worldwide. Konovalov et al. (2014) optimized BB emissions of both particulate matter and CO in Siberia by using satellite measurements of AOD and CO columns and found that the ratios of optimal BB emissions of these species were considerably larger (specifically, by factors of 2.2 and 2.9 in the cases of forest and grass fires, respectively) than those based on typical values of emission factors from literature.

The reasons for the persistent discrepancies between measurements and simulations of AOD in regions affected by large fires remain largely unknown. One the one hand, there is evidence (e.g., Petrenko et al., 2012) that the underestimation of AOD by models can indeed be explained (at least partly) by uncertainties in emission inventories. On the other hand, Konovalov et al. (2015) found (for the case of BB plumes from the Russian 2010 fires) that the agreement between satellite AOD observations and the corresponding simulations constrained with ground based measurements in a source region could

be substantially improved by taking into account the source of secondary organic aerosol (SOA) due to oxidation of semi-volatile organic compounds in the simulations. Although such a source of SOA has been shown to play an important role in the evolution of atmospheric OA originating from both biomass and fossil fuel (see, e.g., Robinson et al., 2007; Grieshop et al., 2009), it has not been taken into account in most of the models employed for simulations of the BB aerosol evolution.

To distinguish errors in model results due to uncertainties in biomass burning emissions from those associated with

shortcomings of the model description of BB aerosol aging processes, Konovalov et al. (2015) considered $PM_{10}$ and CO ground based measurements (matched to corresponding model data) within the fire region and about thousand kilometers downwind. However, such age-resolved measurement datasets have so far been available only from a relatively small number of case studies and field campaigns (e.g., Yokelson et al., 2009; Akagi et al., 2012; Vakkari et al., 2014) and, as noted above, these data provide rather inconsistent evidence about the effects of aging on BB aerosol properties.

The goal of this study is to investigate the feasibility of deriving the information on BB aerosol aging from satellite measurements of AOD and CO columns. To this end, we analyze the data retrieved from MODIS (Remer et al., 2005) and IASI (Clerbaux et al., 2009) measurements of a major biomass burning event that occurred in Siberia in summer 2012 (Konovalov et al., 2014). To estimate the photochemical age of the observed BB plumes, we use model tracers that are





assumed to have the same sources as the actual BB aerosol contributing to the satellite AOD measurements. Unlike ground-based and dedicated aircraft BB plume measurements, limited in time and space, satellite observations provide data on major biomass burning events practically daily and worldwide. Therefore, satellite data combined with simulations may have the potential to substantially enrich current knowledge of BB aerosol aging in different environments and to provide a useful

way for evaluating and improving representation of atmospheric processing of BB aerosol in chemistry transport and climate models.

This paper is organized as follows. Section 2 describes measurement and model data used in our analysis; it also provides a description of the methods used in this study to estimate the photochemical age of BB aerosol and to derive AOD dynamics associated with BB aerosol ageing from satellite measurements. The results of our analysis are presented in Section 3, which

begins with a preliminary comparison of model and simulated data and continues with a presentation of our estimates and tests concerning the dynamics of BB aerosol aging. The results are summarized and discussed in Section 4. Concluding remarks are provided in Section 5.

## 2 Data and method description

### 2.1 Input datasets derived from satellite measurements

2.1.1 AOD measurements

Our analysis is based on using the retrievals of AOD at 550 nm from MODIS measurements onboard the AQUA and TERRA satellites. The satellites are positioned at sun-synchronous circular orbits. The ground path of AQUA crosses the equator at 13:30 LST (at the ascending node), the equator crossing time of TERRA is about 10:30 LST (at the descending node). The AOD retrievals provided nominally at 10 km × 10 km horizontal resolution were obtained as the Level-2 orbital

granule data from the Level 1 and Atmosphere Archive and Distribution System (LAADS) ftp site (ftp://ladsweb.nascom.nasa.gov/allData/51/). The retrieval algorithm and is described elsewhere (Remer et al., 2005; Levy et al., 2007). Validation studies (e.g., Levy et al., 2010) indicate that the multiplicative and additive errors of MODIS retrieved AOD typically do not exceed ±15% and ±0.05, respectively.

The available valid data (with the quality flag ≥ 1) for AOD at 550 nm were unpacked and projected onto a rectangular 1 by

1 degree grid covering the study region and period (as specified below) with hourly temporal resolution. The data (from either of the two satellites) falling into the same grid cell at the same hour were averaged.

2.1.2 CO columns

Along with AOD data, we used the total CO column amounts retrieved from the measurements by the Infrared Atmospheric Sounding Interferometer (IASI) on board the METOP-A satellite (Clerbaux et al., 2009). The Level-2 CO retrievals were

provided by LATMOS/CNRS and ULB (http://ether.ipsl.jussieu.fr/ether/pubipsl/iasi_CO_uk.jsp). The METOP-A satellite is



positioned on a sun synchronous polar orbit with equator crossing at around 21:30 and 9:30 LST for the ascending and descending nodes, respectively. A nominal spatial resolution of the IASI measurements is about 12 km on the ground, within a swath of about 2×1100 km. The CO columns were retrieved from the cloud-screened measurements in the infrared spectrum centered at 4.7 μm by using the FORLI (Fast Optimal Retrievals on Layers for IASI) algorithm (Hurtmans et al.,

2012). Previous studies (e.g., Turquety et al., 2009; Krol et al., 2013; Konovalov et al., 2014) indicated that the IASI CO retrievals provide useful information on atmospheric CO originating from fires. The data were processed in the same way as in Konovalov et al. (2014). In particular, the selection criterion based on the DOFS (the Degree of Freedom of the Signal) values indicating sensitivity of the IASI measurements in the boundary layer (George et al., 2009) was applied: only those data were used that featured a DOFS value exceeding 1.7. The CO retrievals were projected at hourly resolution to the same

grid as the AOD data.

### 2.1.3 FRP measurements and BB emissions

In addition to the atmospheric composition data outlined above, we used the Fire Radiative Power (FRP) data that, like the AOD data, were retrieved from the MODIS measurements (Kaufman et al., 1998; Justice et al., 2002). The FRP data were used in this study to calculate BB emissions, as explained below. Exactly the same data had been used earlier in Konovalov

et al. (2014); accordingly, the details on their processing can be found therein.

Following Kaiser et al. (2009) and Konovalov et al. (2011; 2014; 2015), the BB emissions for a given species $s$, $E^s(t)$ (g s$^{-1}$ m$^{-2}$), in a given grid cell at the moment $t$ were calculated as follows, assuming that BB rate (BBR) is linearly proportional to FRP:

$$E^s(t) = \Phi_d \sum_l \alpha \beta_l^s \rho_l h_l(t) \,, \tag{1}$$

where $\Phi_d$ (W m$^{-2}$) is the daily mean FRP density, $\alpha$ (g[dry biomass] s$^{-1}$ W$^{-1}$) is the empirical factor referred to below as the FRP-to-BBR conversion factor, $\beta_l^s$ (g [model species] g$^{-1}$[dry biomass]) are the emission factors for a given type, $l$, of the land cover, $\rho_l$ is a fraction of a given land cover type, and $h_l(t)$ is the diurnal variation of FRP density. The $\Phi_d$ and $h_l(t)$ were derived directly from the FRP measurements as explained in Konovalov et al. (2014; 2015). In the simulations performed in this study, the daytime maxima of $h_l(t)$ were of 3.2 for forest fires and of 2.7 for other fires. The emission factor values for

organic carbon (OC), black carbon (BC), CO, NO$_x$, and non-methane volatile organic compounds (VOC) were specified using Andreae and Merlet (2001) and subsequent updates (M.O. Andreae, unpublished data, 2014) and were the same as in Konovalov et al. (2015, see Table 2 therein).

Following Konovalov et al. (2011; 2014; 2015), the FRP-to-BBR conversion factor was evaluated by fitting the AOD and CO columns predicted by our model (see Sect. 2.2 below) to the corresponding measurement data. To do so, it was




convenient to represent α as the product of its "a priori" value, $\alpha_0$ (taken to be $3.68 \times 10^{-4}$ g s$^{-1}$W$^{-1}$ as evaluated by Wooster et al., 2005), and the correction factor, $F_\alpha$:

$$\alpha = \alpha_0 F_\alpha ,\qquad(2)$$

The optimal value of $F_\alpha$ was found by minimizing the following cost function, $J$:

$$J = \sum_{j=1}^{Nd} \sum_{i=1}^{Nc} \theta^{ij} \left( V_r^{ij} - V_m^{ij} \right)^2 \qquad(3)$$

where $V_r^{ij}$ are the satellite retrievals of AOD or CO columns, $V_m^{ij}$ are their modeled counterparts, $i$ and $j$ are indices of a grid cell and a day, respectively, and $\theta^{ij}$ is the data selection operator equal either to one or zero. The criteria for the data selection are specified below in Sect. 2.2.2, where it is also explained how the daily values involved in Eq. (3) were calculated using the corresponding hourly values. The minimization procedure was identical to that described in Konovalov et al. (2015). Briefly, it involved "twin" model runs with $F_\alpha$ =0 and $F_\alpha$ =1 and a linear scaling of the difference between their results. When necessary, the estimation was re-iterated using $F_\alpha$ inferred from the previous twin runs.

The optimization of the BB emissions was performed using only data available for the study region that covered the western and central parts of Siberia (50-76°N, 60-111°E) and for the study period that included 46 days of summer 2012, from July 1 to August 15. The study period covered the major fire events that occurred in western and central Siberia during the hot and dry summer of 2012. Fig. 1 illustrates the BB emission sources in the study region by showing the spatial distribution of BBR (which is numerically equivalent to $E^s$ evaluated using Eq. (1) with $\beta_i^s$ equal unity) obtained by averaging "a priori" hourly BBR values (that is, estimated with $\alpha=\alpha_0$) over the study period. Note that the study region covers only the part of the model domain (also shown in Fig. 1) employed for the simulations used in this study. BBR values presented in Fig. 1 were aggregated separately over the two categories of vegetation land cover, the first of which included coniferous and deciduous forest (see Fig. 1a) and the second all other vegetation land cover types, such as (mainly) grass, agricultural land, and shrubs (see Fig. 1b). Our calculations indicate that the forest fires were predominant. Specifically, we estimated that the total amount of the biomass burned in forest fires in the study region and period was a factor of 2.4 larger than that of the biomass burned in other vegetation fires.

### 2.2 Simulated data

2.2.1 The CHIMERE chemistry transport model: brief description and configuration of model runs

The simulations used in the analysis presented below were performed with the CHIMERE chemistry transport model (CTM). The CHIMERE CTM is an Eulerian 3D model that enables simulations of air pollution at different scales, from urban to continental. A detailed description of the model can be found in Menut et al. (2013) as well as in the model documentation (available at http://www.lmd.polytechnique.fr/chimere/ ). CHIMERE was successfully used in a number of





studies including those focusing on environmental effects of wildfires (e.g., Hodzic et al., 2007; Konovalov et al., 2012; Berezin et al., 2013; Péré et al., 2014; Turquety et al., 2014).

The configuration of the simulations performed in this study was largely the same as in Konovalov et al. (2015). One difference with Konovalov et al. (2015) is that the simulations used in this study were done with a lower horizontal

resolution (1°× 1° degree resolution is used instead of the former 0.5°× 0.5° resolution, because a much larger model domain is used in this study). Another difference is that here we used the EDGAR v4.2 data (EC-JRC/PBL, 2011) for anthropogenic emissions (instead of the EMEP emission data that do not cover Siberia). Note that the same anthropogenic emission data were used in the simulations presented by Konovalov et al. (2014). The simulations were performed with 12 layers in the vertical (up to the 200 hPa pressure level). The lateral and upper boundary conditions were specified using the data of

climatological runs of the LMDz-INCA global model (Folberth et al., 2006). Meteorological data were provided by the WRF-ARW (v.3.6) model (Skamarock et al., 2008), which was driven with the NCEP Reanalysis-2 data. A noteworthy feature of our simulations is the use of AOD data retrieved from the MODIS measurements (see Sect. 2.1.1) to account for the effect of BB aerosol on photolysis rates by means of off-line calculations with the TUV model (Madronich et al., 1998), as described in Konovalov et al. (2011); this allowed us to obtain more realistic estimates of the BB aerosol photochemical

age (depending on the OH level that is strongly affected by photolysis reactions) and also to take into account a possible feedback on BB aerosol through the modulation of the OH level and the SOA formation (Konovalov et al., 2016). The injection heights of the BB emissions were parameterized as a function of FRP using the formulation proposed by Sofiev et al. (2012). The initial date of the simulations was 17 June 2012; the first 14 days were used for the model's spin-up.

In this study, the CHIMERE chemical mechanism was insignificantly extended by introducing four "trace" species, one of

which, $T_0$, was chemically passive, while the others, $T_1$, $T_2$ and $T_3$, reacted only with OH (without consuming it). The reaction rates ($k_{OH}$) were $9\times10^{-12}$, $3\times10^{-11}$ and $3\times10^{-12}$ $s^{-1}cm^3$, respectively. Assuming a typical daytime OH concentration of about $4.5\times10^6$ $cm^{-3}$ (as it is follows from ours simulations) these reaction rates correspond to tracer lifetimes of ~ 7, 2, and 20 hours, respectively. The tracers were used in our analysis to estimate the photochemical age of BB OA (as explained below in Sect. 2.3).

In the analysis that follows, we consider the results of two model runs, in which OA evolution was simulated with two different OA schemes, both of which have been described in detail in earlier publications (Menut et al., 2013; Zhang et al., 2013; Konovalov et al., 2015). The first run, referred below to as the "STN" run, was performed with the aerosol scheme implemented in the standard version of CHIMERE (Menut et al., 2013). The aerosol scheme used in the second ("VBS") run was based on the volatility basis set (VBS) method (Donahue et al., 2006; Robinson et al., 2007). In this study, parameter

configurations of the first and second schemes were the same as in the model runs "STN" and "VBS-3" in Konovalov et al. (2015), respectively. The BB aerosol emissions were calibrated independently for each model run (see Sect. 2.1.3). The emission rates for the tracers were specified to be the same as the BB aerosol emission rates in the STN run. The tracer concentrations used in the analysis presented below were simulated in the STN model run. An additional model run (labeled



below as "BGR") was done without BB emissions of aerosols and gases to simulate anthropogenic and biogenic aerosols under "background" (with respect to the BB events) conditions (see Konovalov et al., 2015 for further details).

Note that the OA simulations performed in this study with the two different schemes were only intended to demonstrate the influence of different representations of OA aging processes on the simulated evolution of BB aerosol. Achieving good

quantitative agreement between simulations and measurements (e.g., through optimization of model parameters) in the complex situation considered was beyond the scope of this study. Key assumptions underlying the first ("standard") scheme are that primary OA is composed of non-volatile material and that secondary OA can be formed in a single step from oxidation products of only aromatic and biogenic VOC precursors (Pun et al., 2006). A potentially more important SOA source (Robinson et al., 2007; Grieshop et al., 2009) associated with continuous oxidation of emitted semi-volatile organic

compounds (SVOCs) is disregarded. The second scheme (Zhang et al., 2013; Shrivastava et al., 2013; Konovalov et al., 2015) allows taking into account the well-established fact that BB OA consists of organic compounds covering a broad spectrum of volatilities (see, e.g., May et al., 2013). This latter scheme (referred below to as the "VBS" scheme) also involves representation of SOA formation due to oxidation (functionalization) of SVOCs and IVOCs, as well as an inverse (fragmentation) process resulting in an eventual loss of SOA mass and formation of non-volatile SOA. Similar to Konovalov

et al. (2015), we assumed that BB aerosol was externally mixed with other types of aerosol in the study region. So, to simulate the evolution of BB aerosol (in the STN or VBS runs), we zeroed emissions of other types of aerosol and disabled secondary aerosol formation from anthropogenic and biogenic precursors. Note that, in the same runs, anthropogenic and biogenic emissions of gases as well as the standard boundary conditions for gases were taken into account along with BB emissions of gases in order to ensure a realistic level of oxidant concentrations. The OA and BC emissions were distributed

among 9 size bins according to a lognormal size distribution with a mass mean diameter of 0.25 μm and a geometric standard deviation of 1.6. Similar to Konovalov et al. (2015), an inorganic fraction of BB aerosol that usually provides a minor contribution to BB aerosol mass concentration (Reid et al., 2005a) was disregarded to simplify our analysis. We also disregarded a coarse fraction of BB aerosol that is unlikely to contribute significantly to aerosol properties determining AOD at 550 nm (Reid et al. 2005b).

2.2.2 Using model output data to calculate AOD and CO columns

AOD was evaluated using model output data for mass concentration of aerosol (or aerosol tracers) in the same way as in Konovalov et al. (2014, 2015) following a simple but robust method (Ichoku and Kaufman, 2005) that linearly relates AOD to the aerosol mass column concentration through the mass extinction efficiency (MEE). The latter was evaluated as the sum of the mass scattering efficiency and mass absorption efficiency by taking into account in-situ experimental data collected in

extratropical forests during several campaigns and reviewed by Reid et al. (2005b). Specifically, we assumed, for definiteness, that MEE for BB aerosol equals 4.3 $m^2\ g^{-1}$. Note that according to Reid et al. (2005b) this value (given with an uncertainty of $\pm 0.7\ m^2\ g^{-1}$) best characterizes fresh and dry BB aerosol, whereas the MEE of aged dry BB aerosol tends to be





larger (4.7 ±0.7 $m^2$ $g^{-1}$), but still within the error bars of the value for fresh aerosol. Using a constant value of MEE also implies that the effects of smoke particle hygroscopicity on AOD were also disregarded; possible implications of this simplification for the results of this study are discussed in Sect. 4. The same value of MEE (4.3 $m^2$ $g^{-1}$) was applied to the total mass concentration of other types of aerosol (such as anthropogenic, biogenic and dust aerosols providing only a minor

contribution to the total aerosol mass in the region and period considered), due to the lack of reliable knowledge about their optical properties. A bias in the simulated "background" AOD values that can be caused by such a simplistic assumption was evaluated and taken into account in our analysis as explained later in this section.

The CHIMERE output data for mass concentration of aerosol (or aerosol tracers) and for CO concentration were processed consistently (in time and space) with the corresponding satellite retrievals of AOD and CO columns by taking into account

the availability of reliable measurements and, in the case of CO measurements, their vertical sensitivity. Specifically, if any quality-assured measurement for a given grid cell and hour was unavailable, the corresponding simulated data were disregarded. The simulated CO vertical profiles (partial columns) corresponding to each individual CO retrieval were transformed following recommendations by Fortems-Cheiney et al. (2009) and similar to Konovalov et. (2014) into the total CO columns by applying the individual averaging kernels associated with the measurements. The simulated partial columns

above the upper layer of the CHIMERE CTM were taken to be equal to the corresponding a priori values. The CO columns transformed with different averaging kernels but corresponding to the same model grid cell and hour were averaged.

In the following analysis, we consider only those AOD and CO data that fall into a time window from 09:00 to 12:00 LST (thus with a maximum time lag of three hours). Hourly data falling into this time window and corresponding to the same grid cell and day were averaged. Ideally, the AOD and CO measurements should have been taken during the same hour; however,

because of the difference between the TERRA (or, especially, AQUA) and IASI overpassing times, such a strict requirement would considerably limit the amount of data available for our analysis. We also did not require overlapping of pixels of AOD and CO measurements available for a given day, but only demanded that they fell into the same grid cell (otherwise, that grid cell was excluded from a combined dataset for a given day). Inconsistencies between the AOD and CO measurements are not expected to affect significantly the results and conclusions of our study, not only because we consider

large-scale BB plumes covering multiple grid cells of our model, but mostly because CO measurements and simulations play only an auxiliary role in our analysis (as explained in Sect. 2.4).

Biases in the simulated values of CO columns and AOD are assumed to be due to inaccuracies in their background fractions (as the BB fractions are fitted to observations, see Sect. 2.1.3) and are evaluated in the same way as in Konovalov et al. (2014). Briefly, we selected the grid cells and days corresponding to background conditions (without a significant impact from fires) and averaged the selected measurement and simulated data inside of sufficiently large spatial and temporal windows covering 41 and 21 grid cells in west-to-east and south-to-north directions, respectively, and 15 days around a given grid cell and a day. A relative (in the case of AOD) or an absolute (in the case of CO) difference between these averaged measurement and simulated data is applied to the corresponding values of CO columns or AOD simulated in the



"BGR" model run. Only those grid cells and days were considered to be representative of background conditions, where the contribution of the fires to the simulated values of both CO columns and AOD did not exceed 10 percent. Note that a bias evaluated in such a way cannot fully account for differences between the simulated and observed data on the scales that are smaller than the sizes of the averaging window. However, it is expected that these differences are manifested as quasi-

random uncertainties in our simulations of CO columns or AOD, and are thus taken into account in the confidence intervals of our estimates.

### 2.3 Photochemical age estimation

We estimate the photochemical age, $t_a$, of BB aerosol at a given location by using the model tracers described in Sect. 2.2.1:

$$t_a = -t_c \ln(C_{Ti} / C_{T0}) , \qquad (4)$$

where $C_{T0}$ and $C_{Ti}$ are the mass column amounts of the tracers $T_0$ and $T_i$ ($i$=1,2, or 3), and $t_c$ is the constant time scale (defined below). As explained above (see Sect 2.2.1), one of the tracers, $T_0$, was simulated as a chemically passive model species that could be affected (after being emitted into the atmosphere) only by transport processes, while the other tracer, $T_i$, was provided with a chemical sink by reacting with OH with a constant rate, $k_{OH}$. In an ideal box model simulation with a constant OH concentration, $[OH]_c$, the concentration of $T_1$ would be constant, whereas that of $T_2$ would decrease

exponentially with time ($\sim\exp[-k_{OH}[OH]_c t]$). So, $t_a$ would be equal to the actual time of "aging" of $T_1$, if $t_c$ was defined as follows:

$$t_c = (k_{OH}[OH]_c)^{-1} . \qquad (5)$$

In reality, the concentration of OH varies both in time and space. Moreover, because of horizontal wind shear, the model tracers (or actual BB aerosol) may have different sources and different "history" at different altitudes. Nonetheless, we

expect that the photochemical age estimate given by Eq. (4) can allow us to distinguish between fresh and more aged BB plumes, at least in typical situations where most of the tracer (or actual BB aerosol) amounts are concentrated within the boundary layer. In our analysis, $t_c$ was defined as in Eq. (5), with $[OH]_c$ representing a characteristic mean OH concentration in the BB plumes considered. To evaluate $[OH]_c$, we first calculated weighted vertical mean OH concentrations in each grid cell of a horizontal model domain and in each hour of the study period, with the concentration of the inert tracer $T_0$ used as

the weights. Then, the vertical mean hourly OH concentrations were averaged, taking into account only the grid cells and hours selected for our analysis (see Sect. 2.2.2).

Note that several studies (e.g., Kleinman et al. 2008; Hodzic et al. 2010) estimated the photochemical age of aerosol in air pollution plumes by using observed concentrations of $NO_x$ and $NO_y$ as $-\log_{10}(NO_x/NO_y)$. Conceptually, that estimate is similar to ours, as the $NO_x$ concentration is expected to decrease exponentially by reacting with OH, but the main difference

is that our estimate is based on model data. Both kinds of estimates can only qualitatively translate into an actual time of



aerosol evolution, because the OH concentration in air surrounding any given aerosol particle at any given moment is not known, neither in the real atmosphere nor in our simulations performed with an Eulerian model.

**2.4 Analysis of the AOD enhancement ratio**

2.4.1 Basic formulations

To characterize the effects of BB aerosol aging, we considered the evolution of the AOD enhancement ratio (ER), $\gamma_a$, that can be defined, in a general case, as follows:

$$\gamma_a(t_a) = \frac{(\tau(t_a) - \tau_b(t_a))\tau_t(t_a)^{-1}}{(\tau(t_a = t_0) - \tau_b(t = t_0))\tau_t(t = t_0)^{-1}}, \qquad (6)$$

where $\tau$ is the optical depth of an actual mixture of BB and background aerosol at a given location and at a time corresponding the photochemical age $t_a$, $\tau_b$ is the "background" AOD (in the absence of BB aerosol), and $\tau_t$ is the AOD of a

hypothetical BB aerosol passive tracer that has the same sources as the actual BB aerosol but is not affected by any processes except for advective transport and turbulent mixing. By this definition, $\gamma_a$ is unity when the photochemical age gets a value of $t_0$ (which can, in principle, be chosen arbitrarily). A deviation of $\gamma_a$ from unity may be due to a variety of factors, including chemical processes (such as, e.g., oxidation processes leading to SOA formation) affecting the BB aerosol mass concentration, composition and optical properties, absorption (or condensation) and desorption (or evaporation) of particle

material, wet and dry deposition, and coagulation. Note that the meaning of $\gamma_a$ defined by Eq. (6) is similar to that of the NEMR or ER of OA evaluated in field studies and smog chamber experiments (see Introduction), except that $\gamma_a$ is formulated in terms of AOD. In principle, Eq. (6) permits estimating the ER for AOD at any moment by using the AOD derived from satellite measurements (as $\tau$) and simulated values of $\tau_t$ and $\tau_b$. However, because of transport errors and uncertainties in BB emission data, a direct application of Eq. (6) to the available measurement and simulated data is rather

impractical, as estimates of $\gamma_a$ obtained in such a way would be extremely sensitive to measurement and model errors. Accordingly, instead of employing Eq. (6) directly, we retrieved temporal variations (dynamics) of the ER by solving an optimization problem formulated following a data assimilation approach (in a very basic form).

Specifically, we considered $\gamma_a$ as a parameter that allows adjusting a simulation of the tracer's AOD, $\tau_t$, to a BB part of the observed AOD (denoted below as $\tau_o$). A modeled counterpart of $\tau_o$ (denoted below as $\tau_m$) was defined as follows:

$$\tau_m = \gamma_a \eta \tau_t + \tau_b, \qquad (7)$$

where $\tau_t$ and $\tau_b$ represent AOD (see Sect. 2.2.2) values calculated using passive tracer ($T_0$) and background aerosol concentrations, respectively, and $\eta$ is a constant auxiliary factor depending on $t_0$. The data selected for the analysis were split





into several ($N_a$) bins, such that each bin $n$ covers a different range of photochemical ages ($t_a \in [t_a^n - d_a/2, t_a^n + d_a/2]$, where $t_a^n$ is the median value, and $d_a$ is the "width" of the bins). Then $\gamma_a$ could simply be estimated as a function of $t_a^n$ as follows:

$$\gamma_a(t_a^n) = \arg\min \sum_{i,j \in \Lambda n} \left( \tau_o^{ij} - \tau_m^{ij} \right)^2, \quad n \in [1, N_a] \tag{8}$$

where $i$ and $j$ are the indexes of grid cells and days in the region and period considered, $\Lambda_n$ is a selection of the data

corresponding to a given bin $n$, and $\tau_m$ is given by Eq. (7); for definiteness, $\eta$ (involved in Eq. 7) was evaluated such that $\gamma_a$ for the first bin equals unity. Eq. (8) specifies fitting of the simulated AOD to the corresponding observations by the least squares method through optimization of just one parameter, $\gamma_a$, for each bin $n$.

To reduce the effect of model transport errors on our analysis (as discussed later in this section), the contributions of BB to both $\tau_o$ and $\tau_m$ were normalized to the corresponding BB contributions to the measured and modeled CO columns

approximately matching in time and space the corresponding AOD data. Accordingly, Eq. (8) was modified as follows:

$$\gamma_a(t_a^n) = \arg\min \sum_{i,j \in \Lambda n} \left( \frac{\tau_o^{ij} - \tau_b^{ij}}{v_o^{ij} - v_b^{ij}} - \frac{\tau_m^{ij} - \tau_b^{ij}}{v_m^{ij} - v_b^{ij}} \right)^2, \quad n \in [1, N_a] \tag{9}$$

where $v_o$ and $v_m$ are the CO columns retrieved from the IASI measurements and simulated with CHIMERE, respectively, and $v_b$ are the background CO columns.

The rationale behind using Eq. (9) instead of Eq. (8) is illustrated in Fig. 2, demonstrating the relationships between the

photochemical age and AOD associated with a non-reactive aerosol tracer ($\tau_t$) in comparison with a similar relationship between the photochemical age and AOD normalized to BB CO column amounts ($\tau_t(v_m - v_b)^{-1}$). It is clearly seen that $\tau_t$ exhibits much larger variability than $\tau_t(v_m - v_b)^{-1}$. This fact indicates that $\tau_t$ is likely much more strongly affected by model errors than $\tau_t(v_m - v_b)^{-1}$. Moreover, although both quantities tend to decrease with the photochemical age, the decline of the linear fit for $\tau_t$ (reflecting dilution of BB plumes) is much stronger than that for $\tau_t (v_m - v_b)^{-1}$. Potentially, a significant

relationship between passive tracer concentration and photochemical age can result in systematic errors of $\gamma_a$. For example, in the situation where $\tau_t$ decreases with the age, a mismatch of fresher BB plumes in simulations with older BB plumes in the real atmosphere due to model transport errors could result in an erroneous upward tendency of $\gamma_a$. Note that in an ideal situation with perfect measurements and a model where each bin could be reduced to just one data point, $\gamma_a$ estimated with Eq. (8) or Eq. (9) would be identical to that given by Eq. (6).

To ensure robustness of our estimates of $\gamma_a$ with respect to possible uncertainties associated with the contributions of background aerosol to the AOD and to CO columns, we evaluated $\tau_b$ and $v_b$ in two different ways. In the first way, $\tau_b$ and $v_b$ were calculated directly by using simulation data obtained in the "BGR" run and additionally corrected for possible biases as explained in Sect. 2.2.2. In the second way, we estimated $\tau_b$ and $v_b$ by averaging over the corresponding measured data



representing (based on criteria indicated in Sect. 2.2.2) background conditions and falling inside the same moving windows that were used for estimation of the biases; an additional condition was that the selected values of the AOD and CO column amounts representing the background conditions in a given grid cell and day were not to exceed the corresponding values of $\tau_o$ and $\nu_o$ in the same grid cell and day. An advantage of the first approach is that it potentially enables more realistic

representation of the spatial and temporal variability of $\tau_b$ and $\nu_b$ (if the model is sufficiently accurate). On the other hand, estimates obtained in the second approach are less dependent on model data that may be less accurate than their measurement counterparts.

Before applying Eq. (8) or Eq. (9), the measurement and simulation data were "screened" using a criterion discriminating between more and less dense BB plumes. The criterion was based on values of CO columns, rather than on AOD values, as

we presumed that any limitations imposed on the range of AOD values involved in Eq. (8) were more likely to result in artifacts (biases) in the retrieved dependence of $\gamma_a$ on the photochemical age than those on CO data, which play an auxiliary role in our estimation procedure. Specifically, we considered the following relative differences:

$$\delta_o^{co} = \frac{\nu_o - \nu_b}{\nu_b}.$$

(10)

where the background CO columns, $\nu_b$, were predicted by CHIMERE. Values of $\delta_o^{co}$ along with their counterparts, $\delta_m^c$,

involving simulated CO columns ($\nu_m$) instead of $\nu_o$ were calculated for all CO data points except for those characterizing the background conditions (see Sect. 2.2.2) and then were arranged with respect to their magnitudes. A given data point was selected for our analysis only if both $\delta_o^{co}$ and $\delta_m^c$ values exceeded the value of a certain percentile (considered as an internal parameter, $p_{\%}$, of our procedure) of the corresponding distributions.

2.4.2 Evaluation of the confidence intervals

Confidence intervals for our estimates of $\gamma_a$ were evaluated using a bootstrapping technique (Efron, 1993). Specifically, we first calculated the differences involved in the right hand parts of Eq. (8) or (9). These differences represent the misfits between the model and measurement data (or their combination as in Eq. 9). Then, we replaced the measured AOD values (in the case of Eq. 8) or the fraction involving the measured AOD and CO data (in the case of Eq. 9) with the synthetic data

obtained as the sum of their modeled counterparts and the misfits that had been randomly sampled from a set of their values available for each photochemical age bin. The estimation procedure involving different samples of the synthetic data (instead of the measurement data) were iterated 5000 times, and the spread of estimates of $\gamma_a$ for each bin was used to evaluate the confidence intervals for the optimal estimate of the ER in terms of the 95th percentile. Finally, the difference between the ER estimates due to the difference in the values of $\tau_b$ and $\nu_b$ estimated in the two ways explained above was also added to the

confidence interval for each bin of the photochemical age values. Note that estimation of $\gamma_a$ for a given bin is equivalent to





fitting a linear regression line without the intercept term, which allowed us to verify the confidence intervals by using a commercial statistical software.

### 2.4.3 Configuration of the estimation procedure

The results of the estimation procedure described above depend on just two parameters: the width of the photochemical age

bin, $d_a$, and the threshold percentile, $p_\%$, of the distributions of $\delta_o^{co}$ and $\delta_m^c$. In principle, values of these of parameters can be assigned rather arbitrarily. Larger values of $d_a$ would result in a lower resolution of the retrieved dependence of the ER for AOD ($\gamma_a$) on the BB aerosol photochemical age ($t_a$) but, probably, also in a lower uncertainty range for the estimate of $\gamma_a$ for each photochemical age bin (because each bin would include a larger number of data points). An increase in $p_\%$ could be expected to have a two-fold effect on the uncertainty in the estimates of $\gamma_a$. On the one hand, it would reduce the number of

data points available for any bin, and, therefore, it could result in a larger statistical uncertainty in $\gamma_a(t_a^n)$. On the other hand, a larger $p_\%$ also means that more dense BB plumes would be considered, and so the estimates of $\gamma_a$ would likely be less affected by uncertainties in the background values of the CO columns and AOD. An analysis of the dependence of $\gamma_a$ on $t_a$ within the whole range of possible parameters values would be a rather impractical task and goes far beyond the scope of this study. Instead, taking the above considerations into account, we focus our analysis on the dynamics of $\gamma_a$ retrieved only

with some definite (but still rather arbitrarily chosen) parameter values that would allow us to retrieve any kind of non-trivial and yet statistically significant variations of $\gamma_a(t_a^n)$.

To select these parameter values in a semi-automatic way, we introduced an *ad hoc* "cost function", $G(d_t, p_\%)$, that involves, in essence, the "noise to signal ratio" with respect to variations of $\gamma_a(t_a^n)$ and their uncertainties and, at the same time, favors parameter values providing a higher temporal resolution of our estimates. Specifically, $G(d_t, p_\%)$ was defined as follows:

$$G(d_t, p_\%) = \frac{1}{(N_a - 1)} \left[ \sum_{n=2}^{Na} (\Delta_n)^2 \left( \sum_{n=2}^{Na} (\gamma_a(t_a^n) - 1)^2 \right)^{-1} \right]^{1/2},  \qquad (11)$$

where $\gamma_a(t_a^n)$ and $\Delta_n$ are the optimal value of the ER for the bin $n$ and its confidence interval, and $N_a$ is the total number (depending on both $d_a$ and $p_\%$) of consecutive bins. In Eq. (11), the "signal" is represented by deviations of the ER at times $t_a^n$ from unity, while the term inversely proportional to $(N_a-1)$ introduces a "penalty" for configurations with a low temporal resolution. So, minimization of this cost function is expected to provide a reasonable compromise between the requirements

of low noise to signal ratio and high temporal resolution of our estimates.

As an additional rather arbitrary setting, the left bound of the first bin ($t_a^1 - d_a/2$) was chosen to be 1 hour (taking into account that the measurement data matched to the simulations were averaged hourly, and therefore, our photochemical age estimates have an intrinsic uncertainty of at least 1 hour). We also required that each of the bins (including the first one) contained at least 10 data points.



Minimization of $G(d_t, p_\%)$ is not an easy task, given the essentially nonlinear nature of this function that may have several local minima. However, we did not really need to optimize it exactly, because (as explained above), the "optimal" values of $d_t$ and $p_\%$ are no more "correct" than any other feasible values of those parameters. Instead, the parameter space of $G(d_t, p_\%)$ was scanned by varying both $d_t$ and $p_\%$ with small constant steps within rather wide intervals. Specifically, $d_t$ was changed

with a step of 0.1 hour from 3.0 to 6.0 hours, and $p_\%$ was varied by a step of 0.01 from 0.70 to 0.95. The fact that $G(d_t, p_\%)$ strongly and steadily increased toward the bounds of the parameter intervals considered indicated that the global minimum of $G(d_t, p_\%)$ was unlikely to be found outside of these intervals.

## 3 Results

### 3.1 Preliminary analysis and comparison and of the measurement and model data

In this section, we present an overview of CO columns and AOD retrieved from satellite measurements and simulated with the CHIMERE CTM. Note that as this paper is not focused on aerosol modeling, model results obtained using different representations of aerosol processes are only briefly discussed below. Fig. 3 shows the daily time series of CO columns and AOD averaged over the study region. The BB emissions were optimized (as explained in Sect. 2.1.3) independently for each configuration of the model runs and for each species (CO or aerosol) considered in our analysis, and the corresponding

estimates of the correction factor ($F_\alpha$) are indicated in the figures. Evidently, when the BB emissions are taken into account, the model reproduces the daily variability of both CO columns and AOD rather well (such that the correlation coefficient exceeds 0.85). Big differences between the AOD values simulated with and without BB emissions indicate that fires provided a major contribution to column mass concentration of aerosol in the troposphere over the study region during most of the period considered.

It is noteworthy that the correction factor estimated using the AOD values from the STN run is considerably larger (by a factor of 2) than that estimated using the CO columns. This result is in line with the findings of previous modeling studies (see Introduction) where BB aerosol emissions calculated with the measured emission factors had to be strongly increased in order to avoid negative biases in AOD simulations. However, when the correction factor is estimated using the AOD values from the VBS run, it turns out to be rather close to the CO measurement based estimate of $F_\alpha$. Consistently with the results

of an earlier study (Konovalov et al., 2015), this finding indicates that BB emission estimates derived from AOD measurements by using an inverse modeling approach can considerably depend on the model representation of OA processes.

It may also be noteworthy (and even puzzling) that in spite of the quite noticeable differences between the optimal estimates of $F_\alpha$ for the different aerosol simulations, the corresponding time series of daily AOD values are very similar. That implies

that a comparative analysis of the time series of the mean modeled and measured AOD values does not allow us to tell which of the model configurations is more adequate, in spite of the major differences in the corresponding representations of





aerosol processes. Most likely, the striking similarity of AOD values from the STN and VBS runs is a consequence of the spatial averaging (both in the horizontal and vertical dimensions) of the model output data. Indeed, a comparison of the time series (not shown here) representing simulated data for individual model grid cells reveals quite large differences between temporal variations of aerosol mass concentration predicted by those two runs. Presumably, such quasi-random differences

tend to cancel out each other when the model data are averaged over many grid cells. In contrast, and as expected, the differences between the temporal variations of aerosol mass concentrations from the STN run and the corresponding temporal variations of the aerosol tracer are typically small, even when the data for individual grid cells are considered, except for very rare (in the region and period considered) locations and moments where and when the aerosol concentration simulated in the STN run was affected by wet deposition.

Fig. 4 shows the spatial distributions of the CO columns and AOD averaged in time over the study period. Strong enhancements in the CO columns and AOD due to fires are evident in the central part of the study region, both in the measurements and the simulations performed with BB emissions. Big differences between the simulations made with and without fire emissions indicate that the atmospheric composition was significantly affected by fires over most of the study region. Note that only the simulations performed with the "STN" and "BGR" configurations of CHIMERE are presented in

Fig. 4; the AOD distribution for the VBS run was found to be quite similar to that depicted in Fig. 4d. The model tends to underestimate both CO columns and AOD in the hot spots, but, in general, the simulated and measured distributions are rather similar. These similarities further confirm that our aerosol and CO simulations are rather adequate.

### 3.2 The ER dynamics: measurement-based vs. model estimates

The dynamics of the ER for AOD ($\gamma_a$) estimated according to Eq. (9) is presented in Fig. 5, which shows the results obtained

using the three different aging (reactive) tracers ($T_1$-$T_3$). The ER values inferred from the satellite data are shown in comparison with those derived from the corresponding modeled data (from the STN and VBS model runs). Note that to evaluate the behavior of the ER for the modeled data, values of $\tau_o$ and $\nu_o$ in Eq. (9) were replaced with their modeled counterparts. As an indicator of the "density" of the corresponding BB plumes, Fig. 5 also shows the near-surface $PM_{2.5}$ concentration calculated for each bin by averaging the corresponding output data of the STN run. The parameters of the

estimation procedure were evaluated for each tracer using Eq. (11), as explained in Sect. 2.4, and the parameter values are also indicated in Fig. 5. The changes of the mean OH concentration ($\sim4.5\cdot10^6$ cm$^{-3}$) between the cases considered in Fig. 5 were found to be very insignificant.

Our major finding is that the ER evaluated with the satellite data exhibits statistically significant changes in dependence on the photochemical age. Specifically, it increases (in the case with the tracer $T_1$, see Fig. 5a) from unity at the first bin ($t_a$

$\in[1.0;6.1]$ h) up to 1.9 at the third bin ($t_a\in[11.2;16.3]$), and tends to decrease afterwards. The same type of the ER behavior is evident for the cases with the other traces, $T_2$ (see Fig. 5b) and $T_3$ (see Fig. 5c), although there are quantitative differences between the cases. Taking into account the confidence intervals for our estimates of $\gamma_a$, one can argue that the observed



increase of the ER with the increase of the photochemical age is not an artifact of uncertainties in the measured or simulated data. In contrast, although the decreasing part of the dependence of $\gamma_a$ is quite pronounced, we cannot claim that the decrease of $\gamma_a$ is real rather than being an artifact of uncertainties, e.g., in the assumed background fractions of $\tau_o$ and $\nu_o$. To make a reliable conclusion as to whether or not the ER is actually decreasing, we would have to take into account that the

uncertainties in different bins may be not fully independent. This kind of a statistical analysis goes beyond the scope of the present study. So, in the analysis and discussion that follow we will focus on the increasing part of the dependence of $\gamma_a$ on $t_a$.

Considerable differences between the ER estimates obtained with different aging tracers are an expected consequence of the fact that any grid cell always contains a mixture of aerosol fractions of different origin and age, and so the same AOD (and CO) data can be attributed to different photochemical age bins, depending on the reactivity of the aging tracer. Indeed, the

different fractions get different weight in the photochemical age estimates obtained with different tracers. For example, relatively fresh BB aerosol can be expected to affect a photochemical age estimate obtained with a more reactive tracer (such as $T_2$) much more strongly than a similar estimate obtained with a less reactive tracer (e.g., $T_3$), because the contribution of more aged aerosol fractions to the concentration of $T_3$ at a given moment of time is, by definition, larger than the contribution of the same fractions to the concentration of $T_2$. As a result, the photochemical age estimated with the tracer $T_2$

should typically be smaller than that estimated with the tracer $T_3$. This is indeed evident in Fig. 5. On the other hand, an entrainment of relatively aged BB aerosol into a fresh BB plume would result in a positive shift of the photochemical age estimate; this shift has to be larger with a less reactive tracer. Therefore, it seems reasonable to suggest that if we are interested in distinguishing the effects of BB aerosol aging that presumably occurs on a certain time scale, $T_a$, then, on the one hand, the lifetime of the aging tracer should not be much larger than $T_a$, so that a contribution of "old" aerosol to the

photochemical age estimate could be filtered out. On the other hand, it should not be much smaller than $T_a$ in order to limit an effect of a fresh aerosol fraction on the photochemical age characterizing a more aged aerosol fraction of interest.

A comparison of the results obtained with the different tracers (see Fig. 5) confirms this reasoning: the use of the tracer $T_1$ having a characteristic lifetime of about 7 hours reveals more pronounced or more significant changes on the time scale $T_a$ of the order of 10 hours than the use of the two other tracers having too large or too small lifetimes compared to $T_a$. It would

hardly be reasonable to try to "fit" a tracer's lifetime to $T_a$, because the actual lifetime of any tracer varies strongly both in space and time, reflecting variability of OH. It would be more reasonable to find an "optimal" tracer by using a criterion similar to that based on Eq. (11). In fact, the magnitude of $G(d_t, p_\%)$ with the parameter values indicated in Fig. 5 is found to be indeed smaller in the case with the tracer $T_1$ (0.14) than in the cases with the tracer $T_2$ (0.18) or $T_3$ (0.22). However, formal optimization of a tracer's reactivity goes far beyond the scope of this study.

Another important finding is that the use of the photochemical age estimates allows one to "visualize" the major difference between the BB OA dynamics simulated with the different aerosol schemes. Specifically, whereas the BB aerosol simulated with the standard scheme behaves (as expected) very similar to a passive tracer (the ER deviates from unity only slightly, initially increasing probably due to SOA formation from "traditional" precursors but then slowly decreasing as a result of





deposition), the ER for aerosol simulated with the VBS scheme strongly increases by a factor of 1.9. Furthermore, the analysis presented in Fig. 5 clearly indicates that the VBS scheme enables more adequate representation of BB aerosol dynamics than the standard scheme at the first (growing) stage of BB aerosol aging. Such a conclusion would be impossible to make simply by comparing the time series discussed in Sect. 3.1.

The agreement between the ER estimates derived from the measurements and VBS simulations is obviously not perfect. In particular, the modeled values obtained with the VBS run increase more slowly during the initial stage of the BB aerosol evolution and do not decrease considerably afterwards; moreover, the confidence intervals for the measurement-based and model estimates of the ER do not intersect in the third bin in the case shown in Fig. 5a. Although a likely increase (not taken into account in our simulations) of the mass extinction efficiency by $\sim 10\ \%$ (Reid et al., 2005b) as a result of aerosol aging

would be sufficient to ensure that the difference between the measurement-based and model estimates of the ER is not statistically significant, it could still not bring the measurement-based and modeled estimates of the ER into perfect agreement. These observations indicate that the representation of BB aerosol with the available VBS scheme is not yet fully adequate, and probably could be improved further by optimization of the parameters of the VBS scheme. This task can be addressed in future studies. In particular, other formulations of the VBS scheme are possible (e.g., Shrivastava et al., 2015),

in which fragmentation reactions get larger weight, leading to a net decrease of BB aerosol concentration after some time.

Note that the measurement and model data covered by different photochemical age bins do not necessarily represent the same BB plumes (but at different stages of their evolution). However, we assume that the difference between our ER estimates for different bins is indeed representative of the difference between the optimal ER values for an ensemble of the plumes at different stages of their evolution. An analysis of a particular situation in which it turned out to be possible to trace

the atmospheric evolution of the same BB plumes as they are transported from one ("source") region into another ("receptor") region is presented in the Supplementary material. The analysis reveals that, in line with the results of the more general analysis presented in this section (see Fig. 5a), the optical depth of the actual BB aerosol increased with respect to the optical depth of the BB aerosol simulated with the standard OA scheme as a result of BB aerosol aging.

Fig. 6 illustrates the effect of the data selection procedure (see Eq. (11) and the corresponding explanations in Sect. 2.4) on

the results of our analysis. Specifically, it shows the dependence of the maximum value of the ER on the value of the parameter $p_\%$ (a threshold percentile that was used to select the data (see Sect. 2.4) corresponding to sufficiently dense BB plumes).The dependence was obtained with a fixed width of the photochemical age bin of 5.1 hours, while the photochemical age itself was estimated with the tracer $T_1$ (as in the analysis illustrated in Fig. 5a, where $p_\%$ was equal to 83%). By definition, larger values of $p_\%$ correspond to denser BB plumes: this is evident in the dependence (also shown in

Fig. 6) of the mean near surface concentration of $PM_{2.5}$ in grid cells and days covered by a photochemical age bin in which the ER reaches a maximum value.

The obtained dependence of the maximum ER values on $p_\%$ demonstrates a clear upward trend that extends up to a value of $p_\%$ of 86 %. The fact that the trend is "submerged" in the confidence intervals of individual estimates of the ER does not



necessarily mean that the trend is insignificant, particularly because the uncertainties of the individual estimates are not independent (since the datasets selected for the different estimates were overlapping). Nonetheless, we cannot claim with confidence that the trend is "real" (i.e., is not an artifact of our analysis) either, since our estimates of the ER can be biased to a different extent with different values of $p_\%$. Specifically, apart from the "vertical" uncertainty reflecting discrepancies

between model and measurement data covered by a given photochemical bin, there can also be a "horizontal" uncertainty in our estimates due to mismatches between the photochemical ages of the actual BB aerosol and its simulated counterpart. The mismatches (or, in other words, uncertainties in our estimates of the photochemical age) could occur due to model transport errors, as well as due to uncertainties in modeled spatial-temporal distributions of BB aerosol and CO sources. It seems reasonable to expect that the uncertainties in the photochemical age estimates have a quasi-random character (at least when

the region and period considered are sufficiently large) and are manifested as the "diffusion" of data points forming the dependencies shown in Fig. 6 across different bins. If so, the horizontal uncertainty in our estimates of the ER should be associated with flattening of the dependence of $\gamma_a$ on $t_a$ and with a negative bias in the ER maximum (for given values of $p_\%$ and $d_t$). Such a bias can be different for different selections of the modeled and measured data (and, in particular, can be dependent on $p_\%$), but it is difficult to evaluate without using any independent data on the photochemical age of the real BB

aerosol.

Compared to the maximum ER estimates obtained with the measurement data, the corresponding estimates based on the simulated data do not show any prominent trend. In principle, an upward trend in the dependence of the maximum ER on $p_\%$ can be explained by suggesting that gas-phase oxidation of organic material originating from fires results in formation of, predominantly, rather volatile products that remain mostly in the gas phase unless the density of OA becomes sufficiently

high. The fact that such a trend is missing in the estimates based on the modeled data that were obtained with the VBS scheme may be considered as one more indication that the representation of BB aerosol with the VBS scheme used in this study is not sufficiently adequate. Making a firmer conclusion in this regard is presently not possible in view of the possible bias in our estimates, which has been mentioned above.

### 3.3 Sensitivity tests and Monte Carlo experiments

A goal of the analysis presented in this section is to evaluate the robustness of our major findings presented in the previous section in view of possible uncertainties and biases in the input data. The results of the analysis are presented in Fig. 7.

First (Fig. 7a), we examined to which extent our estimates of the ER could be affected if our estimation procedure would not involve normalizing AOD to CO column amounts, that is, if the estimation was based on Eq. (8) instead of Eq. (9). Indeed, although using the CO columns in Eq. (9) is likely to result in reducing possible uncertainties due to mismatches between

BB plumes in the real atmosphere and in the simulations (as discussed in Sect. 2.4), it might have led to some artifacts in the ER retrievals if there were major systematic discrepancies between the CO columns derived from satellite measurements and simulated with a CTM. The discrepancies may be, e.g., due to model errors in the injection height of BB emissions, but also





due to probable underestimation of CO columns in the IASI retrievals close to emissions (Turquety et al., 2009). However, the test estimation performed with the same parameters as in Fig. 5a but without using the CO data yields almost the same maximum increase (by a factor of 2.0) of the ER as in the base case (see Fig. 5a), although, as expected, the uncertainty of the ER estimates has increased (by ~40% on the average). Moreover, when CO columns were processed instead of AOD

values (again using Eq. 8), the corresponding estimates of $\gamma_a$ (see a purple line in Fig. 7a) were not found to manifest any statistically significant deviations from unity (except for the ER estimate for the last bin, with $t_a$ larger than 26.5 h), as would be expected for the behavior of ER for a passive tracer. Note that underestimation of the retrieved CO columns in fresh BB plumes (corresponding to the first bin of the photochemical age) with respect to those in more aged plumes could result in underestimation of the ER values obtained with Eq. (9) for the other bins, and, therefore, could not explain the increasing

part of the dependence of the ER on the photochemical age (see, e.g., Fig. 5a). Moreover, unlike near-surface $PM_{2.5}$ concentration, CO columns do not exhibit any strong variations across photochemical age bins (see green dashed line in Fig. 7a); this indicates that variability of the ER for CO columns (see purple line in Fig.7a) is likely mostly due to model errors in the vertical distribution of CO. Accordingly, the test results presented in Fig. 7a provide strong evidence that the increasing part of the ER dependence on the photochemical age is not an artifact of any systematic errors in the modeled or simulated

CO columns.

Second (see Fig. 7b), we tried to assess whether or not the increase of the ER can be an artifact of uncertainties in our model estimates of the photochemical age due to transport errors. To roughly evaluate the effect of such errors, we considered how good the model's prediction is that the CO column amount in a given grid cell and on a given day is noticeably (at least by 10 % with the respect to its background value predicted by the model) affected by fires. Specifically, we evaluated the "hit rate",

$h$, defined in our case as follows:

$$h = \frac{n_c}{n_t} , \qquad (12)$$

where $n_c$ is the number of correct predictions (that is, the number of grid cells/days where both the model and measurements indicate that the contribution of fires to a CO column was noticeable), and $n_t$ is the total number of grid cell/days affected by fires according to the measurements. The hit rate was evaluated for each photochemical age bin separately using the original

set of CO columns retrieved from the IASI satellite measurements and those simulated by our model. The values of $h$ (see black crosses in Fig. 7b) were then used to set up Monte Carlo experiments in which the values of AOD and CO columns from the measurements were replaced with synthetic data obtained by mixing the corresponding modeled values in the horizontal plane. That is, the aerosol and CO data for a given grid cell $(i,j)$ were replaced with the data from another grid cell $(k,l)$ (and vice versa). The differences between $k$ and $i$ or $l$ and $j$ were assigned as random values sampled from a Gaussian

distribution $N(0,\sigma_s)$, with the $\sigma_s$ value of 8.0. The value of $\sigma_s$ was defined such that the hit rates calculated for each photochemical age bin with the mixed synthetic data (see green crosses in Fig. 7b) were smaller than the corresponding hit rates calculated with the original data. The experiments were performed both without and with using CO columns for





normalization of AOD values, that is, using the cost function defined by both Eq. (8) and Eq. (9). The AOD values were calculated using the simulated data for the tracer $T_0$.

The results of the corresponding Monte Carlo experiments are presented by solid and dashed lines in Fig. 7b. Evidently, deviations of the ER estimates from unity due to the random replacements of the modeled data turned out to be much smaller

than those found with real data. In the case where CO columns were not used, the maximum deviation was about 31%; there is also an upward trend (starting from the second bin) that can be explained by factors mentioned in Sect. 2.4. The deviations were found to be smallest (less than 12%) in the case where AOD values had been normalized to CO columns according to Eq. (9). Note that the hit rates considered can be smaller than unity not only because of model transport errors or similar errors due to misplacements of BB aerosol emission sources, but also due to many other kinds of uncertainties in the

measured and simulated data. That is, our Monte Carlo experiments are likely to provide an upper limit for the effects of transport errors. Therefore, the results shown in Fig. 7b strongly suggest that the major features of the ER behavior found in the analysis of the real data (see, e.g., Fig. 5a) is unlikely due to model transport errors and associated errors in our estimates of the BB aerosol photochemical age.

Third (Fig. 7c), we investigated whether the derived increase of the ER was a general property of BB aerosol evolution in the

study region, or was mostly associated with a single specific episode, like the one considered in the Supplementary material. To do so, we split the original dataset into two parts, such that the first part included only odd days of the study period, and the second part included only even days. This splitting allowed us to get ER estimates based on entirely different subsets of the data. To enable a straightforward comparison of the ER estimates derived from each data subset, they were obtained for the same photochemical age bins as the base case estimates shown in Fig. 5a; that is, the estimation procedure involved Eq.

(9), but Eq. (11) was not employed in this test.

As a result, we found that the ER estimates derived from the both data subsets demonstrate qualitatively similar "humped" dependencies on the BB aerosol photochemical age, although there are considerable (but not statistically significant) quantitative differences between them. Not surprisingly, withdrawing half of the data from the original dataset resulted in an increase of uncertainties in the ER estimates. Similar results (not shown here) were obtained when the original dataset was

split into two parts in about the middle of the study period, such that the amount of data in the two data subsets was similar. All these results clearly indicate that the increase of the ER is not due to a haphazard feature of a single BB plume observed from satellites, but a rather general property of BB aerosol evolution in the region and period considered.

Fourth (Fig. 7d), we examined the sensitivity of our estimates to possible systematic errors in the injection height of the BB emissions. To do so, we performed a test model run (for the STN scenario) in which the injection height of the BB emissions

was increased: specifically, a factor of two was applied to the injection height estimates calculated in accordance with the method by Sofiev et al. (2012, see Eq. 10 therein). The model data for aerosol tracers and CO from that test run were then used for estimation of the ER instead of the data from the original STN run.



The injection height parameterization implemented in this study can be expected to underestimate the observed height of a part of the major BB plumes reaching the free troposphere (see Sofiev et al., 2012 and Fig. 2d therein). In turn, the underestimation of BB plume heights can result in systematic mismatches between the AOD fields obtained from the observations and from the model in the presence of a strong vertical wind shear. In particular, in such a case, the model is

likely to overestimate AOD for fresh plumes corresponding to the first photochemical age bin, leading to a positive deviation of the ER estimates for other bins from unity, even if the real BB aerosol behaved as a passive tracer. A possible effect of the underestimation of BB plume height on the modeled CO columns is twofold. On one hand, the CO columns can be overestimated for the same reason as AOD, leading to a compensation of a bias associated with the overestimation of the simulated AOD. But on the other hand, the simulated CO columns processed with the IASI averaging kernels can be

underestimated due to decreasing sensitivity of the IASI instrument at lower altitudes, resulting again in a positive bias in the ER estimates. Visual inspection of Fig. 2d in Sofiev et al. (2012) reveals that doubling of the calculated injection height would eliminate its underestimation for most of the cases where the underestimation occurs if the injection height is calculated using the original formula. Therefore, if the deviations from unity of the ER estimates shown in Fig. 5a were actually due to only the underestimation of the injection height of BB emissions, doubling of the injection height values used

in our simulations would result in a major reduction of such deviations and in a decrease of the maximum value of the ER. However, the analysis performed with the output data from the test run (see the orange curve in Fig. 7d) evidently do not reveal any major decrease in the ER maximum, thus indicating that the potential underestimation of the injection height of BB emissions cannot be the primary reason for the increase of the ER in our case.

Finally, in addition to the main test cases presented above, we analyzed some other test cases that will not be discussed here

in detail. In particular, we tried to ensure that the derived evolution of the ER is not an artifact of our data selection procedure (see Eq. 10 and the discussion around it) by using AOD values instead of CO columns in it. We also obtained test estimates of the ER by using the modeled CO columns to which no transformation involving averaging kernels was applied. These modifications of our procedure did not result in any qualitative changes of the ER behavior, either. Accordingly, we came to the conclusion that the strong increase of the ER is not an artifact of our procedure or model errors but, rather, a real

property of BB aerosol evolution in the study region and period.

## 4 Discussion

In this section, we qualitatively discuss possible physical and chemical mechanisms that could be responsible for the distinctive features of BB aerosol evolution, which were revealed in our analysis of the MODIS AOD measurements (see Sect. 3.2 and Fig. 5a). We also discuss possible implications of our results for future studies of BB aerosol.

The ER defined by Eq. (6) and estimated using either Eq. (8) or Eq. (9) can vary with the BB aerosol age as a result of physical and chemical transformations, affecting the optical properties of BB aerosol. These transformations can be initiated by several processes, including nucleation (new particle formation), absorption/evaporation, coagulation and deposition (see,



e.g., Seinfeld and Pandis, 2006), as well as morphological changes (see, e.g., Martin et al., 2013) and heterogeneous oxidation (see, e.g., Kroll et al., 2015). As both nucleation and absorption increase the surface area of the ensemble of aerosol particles, they are likely to result in an increase of AOD and, thus, of the ER, while evaporation and deposition are most likely to lead to the opposite effect. Depending on the aerosol particle concentration and size distribution, coagulation

(which is typically a slow process, except in very dense and fresh plumes) may cause both a decrease of the mass extinction efficiency (and thus AOD) due to a decrease of the number of particles, but possibly also an increase, as the size of the remaining particles grows and they become more efficient in light scattering. Heterogeneous oxidation is likely to lead to a loss of particle carbon to the gas phase as a result of fragmentation reactions (Kroll et al., 2015), although there are also functionalization reactions introducing more oxygen atoms to the particle material.

Substantial changes of the physical and chemical characteristics of BB aerosol have previously been observed in smog chamber experiments when BB emissions were exposed to UV radiation and are believed to be initiated by gas-phase oxidation of SVOCs and IVOCS (e.g., Grieshop et al., 2009; Hennigan et al., 2011; and Heringa et al., 2011). In particular, Grieshop et al. (2009) found that photochemical oxidation of organic gases was associated with an increase of the mass concentration of BB OA by a factor of 1.5 to 2.8 and with an increase of the O:C ratio by a factor of 1.5 (on average). Even

larger changes were detected in the field measurements in southern Africa by Vakkari et al. (2014) who found that atmospheric processing of BB plumes during a few hours resulted not only in a strong increase (up to a factor of 10) of the ratio of BB aerosol and CO mass concentrations but also in a major enhancement (by a factor of 6) of the aerosol scattering coefficient related to CO concentration. A strong increase (by a factor of 2.5) in the aerosol scattering coefficient (normalized to a tracer concentration) was also reported by Akagi et al. (2012) from aircraft measurements of BB plumes in

California.

It seems reasonable to assume that similar changes could have occurred in BB plumes in the study region and that such changes could be responsible for the increase in the ER as a function of the photochemical age. For example, an increase in the scattering coefficient (relative to a tracer concentration) throughout the vertical column of BB aerosol would result in an enhancement of AOD (and thus of the ER), if other optical parameters of the aerosol remained constant. A problem is that

available laboratory and in situ measurements providing evidence for major physical and chemical transformations of BB aerosol address rather short periods of BB evolution (< 5 h). Such periods would be covered by the first bin of our photochemical age estimates, and therefore the results of the experimental studies mentioned above do not have direct implications for the interpretation of our findings. However, as long as some of the primary BB aerosol, which is likely to be composed mainly of SVOCs (see, e.g., May et al., 2013), evaporates upon dilution (which normally does not occur in BB

aerosol aging chamber experiments), there can be a continuous influx of fresh organic material into the gas phase. As a result of gas-phase oxidation, this material is likely to gain in mass, lose in volatility, and be absorbed by BB aerosol particles (Grieshop et al., 2009). Accordingly, it may be suggested that the physical and chemical processes leading to an increase of





AOD can remain efficient on a much longer time scales than those addressed in chamber experiments, depending on the temporal scale of dilution (which, in turn, can vary, depending on meteorological conditions).

Such a scenario is confirmed by our simulations based on the VBS scheme. Indeed, with the reaction rate of SVOCs with OH of $2 \times 10^{-11}$ s$^{-1}$ cm$^3$ (based on the recommendations by Grieshop et al., 2009) and a typical OH concentration of $4.5 \times 10^6$

cm$^{-3}$ in daytime, the gas-phase oxidation of primary organic aerosol material, if it occurred in one step, could be mostly accomplished within 3-4 hours, while actually (see Fig. 5) it takes up to ~ 20 hours (or even longer, depending on the photochemical age estimates). The fact that there are quantitative differences between the ER estimates based on the measurements and the VBS simulations in the growing stage of the BB aerosol evolution (specifically, the ER growth rate is underestimated by the model) is not surprising, taking into account that the OA evolution simulated in the framework of the

VBS method depends on many parameters (Donahue et al., 2006), none of which were optimized for the particular situation considered in this study. In addition, it should be noted that the ER estimates based on the simulated data were obtained under the assumption that the aerosol mass extinction efficiency does not change as the aerosol ages, while the available measurement data (Reid et al., 2005b) indicate that it is usually greater for dry aged BB aerosol than for dry fresh BB aerosol.

Typically, the difference in the mass extinction efficiency for fresh and aged BB aerosol is rather small (~10%) (Reid et al., 2005b). However, much larger changes in the mass extinction efficiency can be expected to occur due to new particle formation, which was observed in fresh BB smoke (with an age of less than 2 hours) both in the real atmosphere (Hobbs et al., 2003; Vakkari et al., 2014) and in the laboratory (Hennigan et al., 2012), but is not taken into account in our model. To the best of our knowledge, there is no experimental evidence that new particle formation can be important after the BB

aerosol age exceeds a few hours. Moreover, the smog chamber experiment by Hennigan et al. (2012) indicates that the particle formation rate drops abruptly after just one hour of BB smoke evolution (that is, after a period that was entirely excluded from our analysis). Accordingly, the discussed behavior of our ER estimates derived from satellite measurements, as well as their difference from the ER estimates obtained from the model data are unlikely to be due to the effects of new particle formation.

Considerable changes in the mass extinction efficiency of BB aerosol in the real atmosphere can also be due to absorption of water by BB aerosol particles. This process is also not taken into account in our simulations. The effect of water uptake on light scattering by aerosol particles can be characterized by the aerosol humid growth factor, $f$(RH) (see, e.g., Day et al., 2006), while the impact of humidification on light absorption by BB aerosol is believed to be negligible (Reid et al., 2005b). The product of the mass scattering efficiency of dry aerosol with $f$(RH) provides an estimate of the mass scattering efficiency

of the aerosol in an ambient atmosphere. The humid growth factor depends on the aerosol particle composition and size distribution, but the key parameter determining its magnitude is ambient RH (Kasten, 1969). The BB plumes considered in this study evolved in a relatively dry atmosphere, where RH was typically lower than 60% in the daytime (when the satellite measurements were taken). Mean RH values calculated as a function of the BB aerosol photochemical age and





corresponding to the same grid cells / hours that were taken into account in our analysis illustrated in Fig. 5a are presented in Fig. 8 along with the corresponding values of temperature. Both RH and temperature were calculated using the WRF-ARW model data (see Sect. 2.2.1) as a weighted vertical average, with the concentrations of the inert tracer $T_0$ used as the weights. In addition, Fig. 8 also shows their mean values for the first (near–surface) model layer. It can be seen, that while the

vertically averaged temperature decreases with the photochemical age (due to transport of aerosol into higher levels), RH remains nearly constant and below 60%. Available measurements (see, e.g., Hand et al., 2010) suggest that $f$(RH) is likely to remain smaller than 1.2 under such conditions. Therefore, even though $f$(RH) might increase with aerosol age due to changes in aerosol composition, aerosol particle humidification can explain neither the major increase in the ER as BB aerosol ages nor the difference between the ER estimates based on measurement and model data.

The decreasing part of the measurement-based dependence of the ER on the photochemical age is more difficult to explain, particularly because it is not reproduced by our model. As noted in Sect. 3.2, we cannot claim that the ER is actually decreasing after the initial growth, taking into account the uncertainty range of our estimates. Nonetheless, as the decreasing part is a general feature of the dependencies presented above (see Fig. 5), it may be not entirely an artifact of our analysis. It can be suggested that the decrease of the ER can reflect a decrease in BB OA mass concentration due to either evaporation of

SOA, followed by fragmentation rather than functionalization reactions of the evaporated material (Donahue et al., 2012; 2013), or due to heterogeneous oxidation (George and Abbatt, 2010; Kroll et al., 2015), which can be a major oxidation mechanism for OA particles over timescales of several days, but is not taken into account in our model, or both. Importantly, the behavior of the ER according to our analysis is qualitatively consistent with the current theoretical understanding of the evolution of OA in the atmosphere: in particular, a similar "humped" dependence of OA concentration was obtained by

Donahue et al. (2013, see Fig. 3 therein) in a simulation with an idealized box model. Furthermore, Donahue et al. (2013) pointed out that real atmospheric OA can, in some cases, be oxidized very quickly (e.g., over less than 1 day), depending on environmental conditions (and OH concentration in particular). Our results indicate (see Fig. 5c) that if the downward trend continues beyond the time scales considered, the ER will eventually become less then unity. In this sense, our analysis does not contradict earlier reports (Jolleys et al. 2012; 2015), which indicated that the normalized excess ratios for aged BB

aerosol measured in several aircraft campaigns were, on average, smaller than for fresh BB aerosol.

The results of our analysis have direct implications for the development and evaluation of models representing BB aerosol evolution. In particular, they confirm the findings of earlier studies (e.g., Hodzich et al., 2010; Konovalov et al., 2015; Shrivastava et al., 2015) indicating the importance of taking BB aerosol transformations associated with the oxidation of SVOCs into account in CTMs. Indeed, the simulations based on the VBS method (which provides a convenient framework

for model representations of processes involving SVOCs) demonstrate a much better overall agreement with the ER estimates derived from satellite measurements than the simulation based on a simpler ("conventional") OA scheme, in which primary BB OA is assumed to be composed of only non-volatile material and in which an SOA source associated with oxidation of SVOCs is not taken into account. Our results also support the argumentation (Konovalov et al., 2015) that



inadequate representation of BB aerosol aging processes may be one of the key reasons for the systematic underestimation of AOD by CTMs employing a conventional OA scheme (see Introduction and references therein). Indeed, on the one hand, our analysis indicates that when the BB aerosol simulations are based on the VBS method, the optimal agreement between the simulated and measured AOD data can be achieved with smaller (by a factor of 1.8 in our case) particulate matter

emissions than when BB aerosol simulations are performed using the conventional aerosol scheme. On the other hand, as noted above, the VBS method is found to provide a more realistic representation of the effects of BB aerosol aging than the conventional aerosol scheme. Future modeling studies should involve optimization of the parameters of the VBS scheme in order to better fit the ER estimates obtained with the model data to those derived from measurements.

Our results may also have implications for experimental studies of BB aerosol evolution. Indeed, our analysis provided
observational evidence (to the best of our knowledge, for the first time) of major BB aerosol transformations that occur in the real atmosphere on temporal scales in the range from ~5 to ~20 hours. This is a rather unexpected finding in view of previous studies of BB aerosol evolution, which focused mostly on either shorter or longer time scales. Therefore, our results indicate that more focus should be given in laboratory and field studies to the time scales corresponding to meso- and synoptic scale transport of BB plumes. It would be useful to pay special attention to the role of dilution and to the
dependence of aging processes on the ambient concentration of BB aerosol, and it would be important to ensure (in field experiments) that the photochemical age of a BB plume is well defined and is not affected significantly by entrainment of fresher or more aged smoke. The BB aerosol photochemical age estimation method proposed in this paper may be helpful in this regard.

Finally, we would like to point out that our analysis based on Eq. (8) or Eq. (9) provides a convenient tool to visualize the
effects of BB aerosol aging in simulated data, as it is done, for example, with simulations performed with the different aerosol schemes (cf. red and brown lines in Fig. 5). We believe that, in this way, our method can be helpful, in particular, in various sensitivity studies, as it allows one to easily observe the effects of a parameter change on BB aerosol evolution. Note that an analysis of the effects associated with aerosol aging could be quite straightforward when using a Lagrangian model. However, with an Eulerian model, a practical alternative to the method described in this paper would be a rather
cumbersome analysis of transport patterns and spatial-temporal distributions of BB aerosol (like the special case analysis presented in the Supplementary material).

## 5 Conclusions

In this paper we used MODIS AOD measurements combined with IASI CO measurements and output data from the CHIMERE CTM to probe the effects of aging of BB aerosol in smoke plumes from Siberian wildfires. To quantify the AOD
changes due to BB aerosol aging, we considered the AOD "enhancement ratio" (ER) that indicates how much the optical depth of actual BB aerosol changes with respect to the optical depth associated with a passive aerosol tracer originating from the same sources as the actual aerosol. The AOD ER has basically the same meaning as the normalized excess mixing ratio





(NEMR) used in earlier studies involving in-situ measurements (e.g., Jolleys et al., 2012; Vakkari et al., 2014; Konovalov et al., 2015), but is defined in terms of AOD values rather than mixing ratios. The satellite-observation-based analysis obviously allows obtaining results for larger regions and for the larger number of BB plumes than in-situ measurements. The ER was evaluated as a function of the BB aerosol photochemical age by matching the measured AOD values to their

modeled counterparts calculated as the AOD of a passive aerosol tracer featuring the same sources as the BB aerosol but not affected by any process except for transport. The photochemical age was estimated by combining the passive aerosol tracer concentration with that of another virtual tracer reacting with OH. The model was also used to evaluate "background" AOD values in the absence of fires. The CO columns derived from the IASI measurements and simulated by the model played an auxiliary role in our analysis, counterbalancing possible mismatches between the measured and simulated AOD fields. The

analysis addressed a big region (50-76°N, 60-111°E) in Central and Western Siberia in summer 2012 (specifically, from 1 July to 15 August) where and when intense and numerous forest fires had occurred.

We found that in sufficiently "dense" plumes (where the near-surface $PM_{2.5}$ concentration is of the order of 100 µg m$^3$), the ER demonstrates a substantial (up to a factor of 1.9) and statistically significant enhancement on the time scale of up to 15 hours. The increase of the ER was found to be followed by a decrease; however, the presence of the decreasing part of the

ER evolution cannot be considered as a well established fact in view of the uncertainties in our estimates. The ER was found to increase less in less dense plumes. The robustness of our finding that real BB aerosol evolves quite distinctly from a hypothetical aerosol tracer on the time scales considered is corroborated by the results of our analysis of several test cases involving different modifications of the input data and estimation procedure, as well as by a Monte Carlo experiment intended to replicate the effects of model transport errors. The formal statistical analysis of the BB aerosol evolution is

supplemented by the analysis of a special case where an enhancement of AOD with respect to the CO columns was associated with transport of BB plumes from a "source" to a "receptor" region.

We also found that the increase in the ER can, to a large extent but not completely, be reproduced by the BB aerosol simulations using the same VBS scheme that was successfully evaluated previously in Konovalov et al. (2015) for the case of BB plumes originating from fires in the European part of Russia. At the same time, no significant enhancement was found

in the ER estimates derived from AOD data simulated with the "standard" aerosol scheme of CHIMERE, in which processes involving SVOCs are disregarded. These results and the fact that the initial BB evolution over the first few hours after emission could not be resolved in our analysis allowed us to suggest that the "observed" enhancement of the ER is particularly due to the oxidation of SVOCs evaporated upon dilution from aerosol particles.

Finally, we discussed numerous implications of our findings for future experimental and modeling studies of BB aerosol. In

particular, our findings emphasize the importance of carefully planned field studies addressing time scales exceeding those (of the order of a few hours) typical for chamber experiments, as well as the importance of a proper representation of physical and chemical processes involving SVOCs in CTMs. We also argued that the analysis method presented in this paper can become a helpful tool for developing and validation of CTMs.





*Acknowledgements.* This study was supported by the Russian Science Foundation (grant agreement no. 15-17-10024). The simulations involved in the analysis were performed with support from Russian Foundation for Basic Research (grant nos. 14-05-00481 and 15-45-02516). I.B. Konovalov acknowledges travel expenses in the framework of the PARCS (Pollution in the ARCtic System - PARCS) national project.

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



**(a)**                                              **(b)**

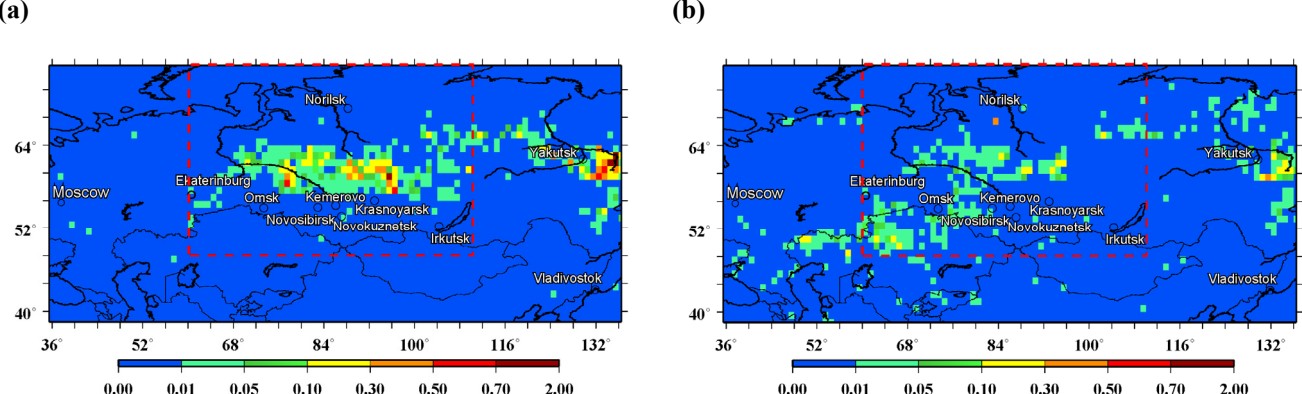

**Figure 1: Mean biomass burning rate (g m$^{-2}$ h$^{-1}$) for (a) forest fires and (b) other vegetation fires in the territories covered by the domain of the CHIMERE model employed in the simulations performed in this study over the period from 1 July to 15 August 2012 according to the calculations based on the MODIS FRP data. Also shown (see red dashed rectangles) is the study region.**

(a)                                                  (b)

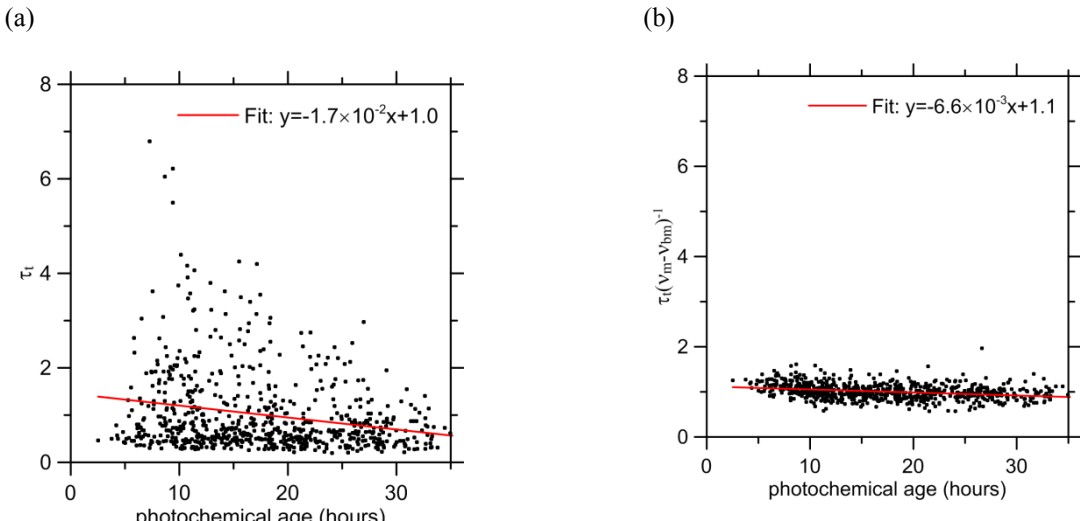

**Figure 2: (a) Relationship between aerosol photochemical age (estimated using the model tracers $T_0$ and $T_1$) and the modeled BB aerosol optical depth ($\tau_t$) calculated by using the simulated concentrations of the passive BB aerosol tracer ($T_o$); (b) the same as in the plot (a) but for $\tau_t$ normalized to the BB fraction of the simulated CO columns ($v_m - v_b$). Different data points correspond to different grid cells and/or days. For better readability, only each 10$^{th}$ data point is depicted. Both $\tau_t$ and $\tau_t (v_m - v_b)^{-1}$ have been normalized to their mean values.**





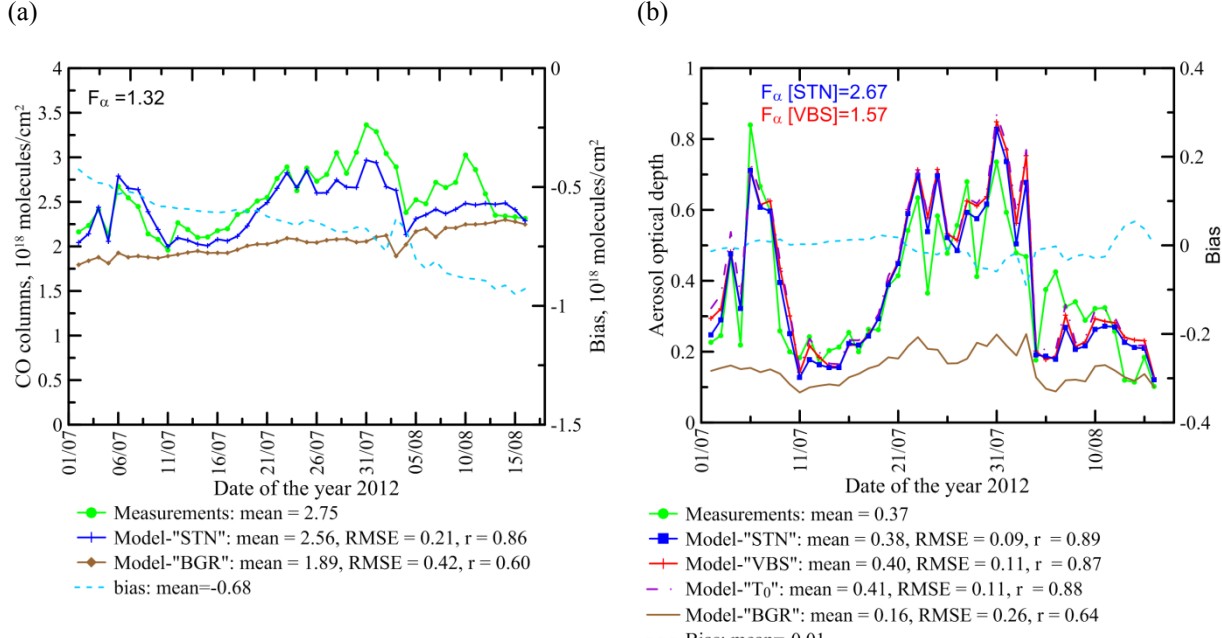

**Figure 3: Time series of daily values of (a) CO columns and (b) AOD averaged over the study region according to IASI CO and MODIS AOD measurements and corresponding simulated (bias-corrected) data obtained in different runs (STN and VBS) of the CHIMERE model and combined with the data from the background (BGR) model run. The BB emissions used in the simulations were optimized (see Sect. 2.1.3) independently for each model run and for each species (CO or aerosol) considered; the corresponding estimates of the correction factors, $F_\alpha$, are indicated in the figures. The modeled AOD values marked as "$T_0$" were obtained using passive tracer ($T_0$) concentrations simulated in the STN run and are almost indistinguishable from those calculated with of "actual" BB aerosol data from the STN run.**

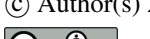



**Figure 4: Spatial distributions of (a, c, e) the CO columns ($10^{18}$ molecules cm$^{-2}$) and (b, d, f) AOD according to (a, b) satellite (IASI and MODIS) measurements and simulations (for the STN case) performed (c, d) with and (e, f) without BB emissions. The CO columns and AOD values shown are averaged over the study period (1 July – 15 August 2012).**





(a)



(b)

(c)

**Figure 5: The AOD enhancement ratio (ER) estimates ($\gamma_a$) as a function of the BB aerosol photochemical age evaluated using different aging tracers: (a) $T_1$, (b) $T_2$, and (c) $T_3$. The OH reaction rates corresponding to each tracer, along with the values of the parameters involved in the analysis (see Sect. 2.4) are indicated in the plots. Along with the ER estimates derived from data of satellite (MODIS and IASI) measurements, the figure shows the estimates obtained by substituting the measurement data with their modeled counterparts provided by the CHIMERE model, which was run with two different aerosol schemes (STN and VBS). Also shown is the mean near-surface $PM_{2.5}$ concentration corresponding to the grid cells and days represented by the data covered by each photochemical age bin.**





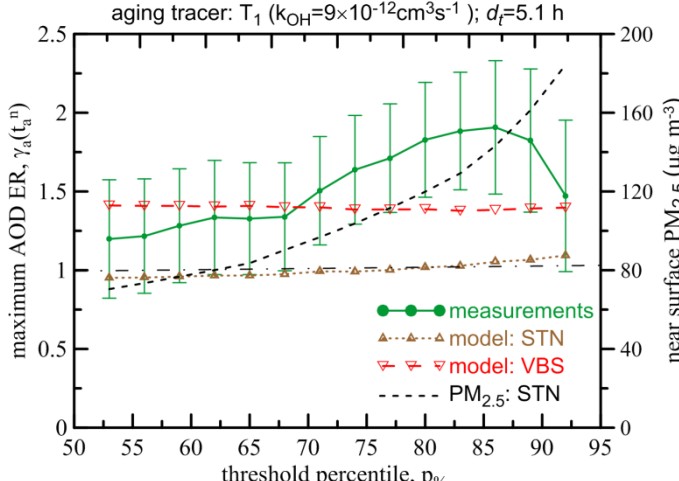

**Figure 6:** The ER maximum values reached in dependence on the photochemical age (see Fig. 5a) as a function of the threshold percentile, $p_\%$, of the distributions of the BB fraction of the CO columns related to their background fraction (see Eq.10). The threshold percentile was used to select the data (see Sect. 2.4) corresponding to sufficiently dense BB plumes. Also shown is the mean near-surface $PM_{2.5}$ concentration corresponding to the grid cells and days represented by the data covered by the photochemical age bin corresponding to an ER maximum.







**Figure 7: Results of the sensitivity tests and Monte Carlo experiments examining the robustness of the estimates presented in Fig. 5a: (a) the ER estimates derived from AOD MODIS measurements without using CO columns for the their normalization, as well as the estimates obtained using satellite measurements of CO columns as proxy for the AOD measurements in Eq. (8); also shown are the mean IASI CO column amounts for corresponding bins; (b) the ER estimates obtained in the Monte Carlo experiments in which the measured AOD values were replaced with the synthetic data generated by means of random replacements of the simulated AOD and CO values (based on the modeled concentration of the passive tracer $T_0$) in the horizontal domain (see experimental details in Sect. 3.3); (c) the ER estimates derived from the two data subsets obtained by splitting the original dataset such that the first and second subsets included only odd and even days of the study period, respectively; note that the "odd days" estimates are depicted by using the upper abscissa axis, which is slightly shifted to improve readability; (d) estimates derived from the original MODIS and CO data but using the data of the CHIMERE run in which the injection heights of BB emissions were doubled.**



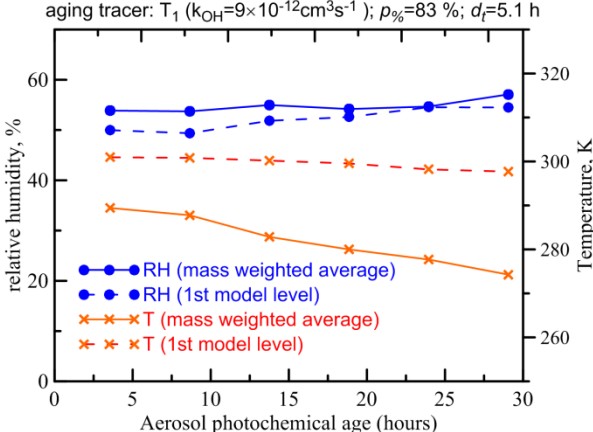

**Figure 8: Mean relative humidity (RH) and temperature values corresponding to the same grid cells / hours that were taken into account in the analysis illustrated in Fig. 5a. Both RH and temperature were calculated using the WRF-ARW model data as a weighted vertical average, with the concentrations of the inert tracer $T_0$ used as the weights. The mean values for the first (near–surface) model layer are also shown.**