# Peer review of "Probing into the aging dynamics of biomass burning aerosol by using satellite measurements of aerosol optical depth and carbon monoxide"

_Atmospheric Chemistry and Physics, 2016_

## Short Comment (SC1) · 25 Oct 2016

This is a short comment on the MODIS data used, not a full review of the paper.

I see that the authors used the Collection 5.1 MODIS data set (based on the ftp link given and papers cited). This version was superseded by the current Collection 6 in 2013 (for Aqua) and 2014 (for Terra), and in general we strongly recommend the use of the latest versions of the data set for scientific analyses. The Collection 6 reprocessing includes the whole missions, not just these latest years.

[Figure]

As well as miscellaneous algorithm updates between the data versions, Collection 6 data make use of an improved level 1 MODIS sensor calibration. This improved calibration has the strongest influence on MODIS Terra, where it decreases the magnitude of artificial trends which were found in the Collection 5.1 data products.

Due to the non-linear response of top-of-atmosphere reflectances to aerosol, calibration updates may have a stronger effect on high-AOD cases such as biomass burning smoke considered in this manuscript.

More information about Collection 5.1 trending issues can be found in Levy et al. (2010), which the authors cite, as well as Lyapustin et al. (2014). The Collection 6 Dark Target land and ocean algorithm paper, with some validation results, is Levy et al. (2013).

Additionally, Collection 6 makes use of an updated land/water mask. This updated mask results in better identification of small inland water bodies, which in turn affects algorithm decisions and quantitative retrieval results. I mention this because high northern latitudes contain many areas with these small lakes and we see changes in retrievals in many of these regions (e.g. Canada, the northern USA, and Siberia). More information about the land mask updates and effects on aerosol products can be found in Carroll et al. (2016).

I would thus strongly encourage the authors, if possible, to make use of the latest versions of the aerosol products.

The Collection 6 aerosol products are available free of charge from the same web site and ftp server where the Collection 5.1 products currently used were obtained. The

differences in file format between the two versions are minimal, and I am happy to help provide advice to the authors about updating to the latest version of the MODIS data set.

References:

M. L. Carroll, C. M. DiMiceli, J. R. G. Townshend, R. A. Sohlberg, A. I. Elders, S. Devadiga, A. M. Sayer R. C. Levy, Development of an operational land water mask for MODIS Collection 6, and influence on downstream data products, International Journal of Digital Earth, doi:10.1080/17538947.2016.1232756, 2016.

Levy, R. C., Mattoo, S., Munchak, L. A., Remer, L. A., Sayer, A. M., Patadia, F., and Hsu, N. C.: The Collection 6 MODIS aerosol products over land and ocean, Atmos. Meas. Tech., 6, 2989-3034, doi:10.5194/amt-6-2989-2013, 2013.

Lyapustin, A., Wang, Y., Xiong, X., Meister, G., Platnick, S., Levy, R., Franz, B., Korkin, S., Hilker, T., Tucker, J., Hall, F., Sellers, P., Wu, A., and Angal, A.: Scientific impact of MODIS C5 calibration degradation and C6+ improvements, Atmos. Meas. Tech., 7, 4353-4365, doi:10.5194/amt-7-4353-2014, 2014.

---

## Short Comment (SC2) · 2 Nov 2016

I thank Dr. Sayer for his interest in our paper. Of course, we have been aware that the MODIS data were being reprocessed and that the Collection 6 has been released. However, when we configured our study in January-February of this year, the Collection 6 AOD data had not yet been made available through the Giovanni interactive visualization and analysis system (http://giovanni.gsfc.nasa.gov/giovanni) operated by NASA. Consequently, on the one hand, we could not be sure that the reprocessing of the MODIS data had been accomplished and the Collection 6 dataset for 2012 had been finalized and verified. On the other hand, we would not be able to ensure (by using the visualization tools and corresponding Level 3 data which are provided by the
Giovanni system) that we processed the Level 2 data properly and our dataset is complete. Nonetheless, by comparing the different versions of the Level 2 data that were then available through ftp, we made sure that the differences between the Collections 5.1 and 6 data projected to our model grid were small compared to the differences between our modeled AOD data and any of the MODIS AOD data, and so we presumed that the differences between the Collection 5.1 and Collection 6 data were unimportant in the context of our study.

To get a better idea about possible effects of the changes in the MODIS data on the results of our study, I downloaded the Collection 6 Level 3 AOD data (now available through Giovanni) for the two days (21 and 22 July) which were analyzed in the Supplementary material for our paper and then combined the Aqua and Terra data together (such that if the data for a given grid cell were available from the both satellites, they were averaged). In the same way, I also combined the Collection 5.1 Level 3 data from Aqua and Terra satellites. Finally, the both data sets were used in the analysis illustrated in Fig. S6 instead of the original dataset (based on the Collection 5.1 Level 2 data) used in our study. The results are shown in the graphical supplement for this comment. It can be seen that although the regression coefficients representing the enhancement ratio for AOD with respect to CO columns are insignificantly smaller (by about 15 percent) in the case with the Collection 5.1 data, the difference between the values of the regression coefficient for the source and "receptor" region is even slightly increased with the Collection 6 data. This means that, qualitatively, the conclusions which can be made for this situation by using the Collection 6 data remain exactly the same as the conclusions that were made with the original data (see the Supplementary material for our manuscript). It is also noteworthy that the differences between our modeled AOD data and any of the MODIS AOD data are typically much larger than between the corresponding Collection 5.1 and Collection 6 data.

Furthermore, by considering all the Level 3 data available for the study region for 22 July, I found that the correlation coefficient between the Collection 6 data and our modeled data remained about the same as in the case with the Collection 5.1 data (∼0.67). So I could not find clear evidence that the use of Collection 6 data would lead to a reduction of uncertainties in our results (although this does not mean that the Collection 6 data are less accurate than the Collection 5.1 data). Taking into account the results presented in Fig. 7a (specifically, the fact that the CO column amounts corresponding to the BB plumes considered do not change significantly with the photochemical age of the plumes), I also do not see how correcting a minor bias in the Collection 5.1 data for dense BB plumes can invalidate our findings indicating major changes in AOD due to BB aerosol ageing, as well as I cannot imagine any solid reason for significant changes in our results due to better identification of small inland water bodies in the MODIS retrievals. On the other hand, I noted that the number of data points provided in the Collection 6 dataset has slightly decreased with respect to the Collection 5.1 dataset; it means that the use of the Collection 6 data could potentially result in larger uncertainties in the results of our analysis.

Accordingly, I'm confident that the major findings of our study cannot be affected by differences between the Collection 5.1 and Collection 6 data. I hope for the understanding that so much as re-processing of satellite data may take years, a complex analysis involving model data cannot be re-done overnight after the release of a new version of satellite retrievals. I think that some transitional period should be allowed in this regard. Note that the MODIS measurements were involved in our calculations of photolysis rates; so in order to replace the Collection 5.l data with the Collection 6 data, we will need to fully re-do not only our analysis but also the underlying simulations with a chemistry transport model. I strongly believe that such a significant effort would not be worthwhile. I believe also that there is common understanding that the fact that the Collection 5.1 MODIS data is now superseded by the current Collection 6 does not invalidate numerous studies in which the Collection 5.1 (and older) data were involved.

Finally, I would like to thank Dr. Sayer for his kind offer of help with using the Collection 6 data. I will be happy to contact him in case any relevant questions arise in our work,

and I hope that this discussion will initiate mutually useful collaboration.

Please also note the supplement to this comment:
http://www.atmos-chem-phys-discuss.net/acp-2016-797/acp-2016-797-SC2-
supplement.pdf

**Supplement:**

The figures shown below present the same analysis as in Fig. S6, except that the Collections 5.1 and 6 Level 3 AOD data were used instead of the AOD dataset based on the Collection 5.1 Level 2 data. Specifically, the figures show the relationships between BB fractions of CO columns ($\Delta v$) and AOD ($\Delta \tau$) in (a) the "source" region on 21 July and (b) in the "receptor" region on 22 July 2012 according to the satellite (MODIS and IASI) measurements (see green and blue lines) and the simulations for the STN scenario (see orange dashed lines) The source and receptor regions are shown in Fig. S1. We consider an increase of the difference between the regression coefficients for the observed and simulated data in the receptor region with respect to the source region as evidence for the enhancement of AOD due to BB aerosol aging processes. It can be seen that this increase is slightly larger (0.45 vs. 0.42) in the case with the Collection 6 data than in the case with the Collection 5.1 data. Accordingly, this analysis indicates that the AOD dynamics discussed in our manuscript is not an artifact of the known minor issues in the Collection 5.1 data.

---

## Short Comment (SC3) · 2 Nov 2016

I thank the authors for the follow-up here. It is good that the Collection 5/6 difference does not appear to be too large for this specific application. If feasible I would personally encourage repetition with the latest version anyway (if the data volume is smaller in Collection 6 than Collection 5 that probably means there were some problematic issues in the Collection 5 pixels which are missing, which resulted in their removal from Collection 6). However I can certainly understand that analyses cannot be repeated overnight.

I wanted to add one note for the future, regarding Giovanni. Giovanni is primarily a visualisation service and not the main official distribution source for MODIS aerosol

or other data products. It typically lags behind the official sources by up to a few years in terms of ingesting newer versions of satellite products (as noted the MODIS Collection 6 for Aqua became available in 2013 - I am not sure when Giovanni made the update), and is not maintained by the science teams responsible for algorithm development/validation.

So for future analyses, we always strongly encourage users to go to the official Distributed Active Archive Centers (DAACs) for NASA data products, which in the case of MODIS aerosols is the LAADS (https://ladsweb.nascom.nasa.gov/). This ensures that they are using the latest versions of data products, and have access to the latest documentation about them.

---

## Referee Comment (RC1) · Anonymous Referee #2 · 23 Nov 2016

General Comments:

The manuscript on "Probing into the aging dynamics of biomass burning aerosol by using satellite measurements of aerosol optical depth and carbon monoxide", by Konovalov et al. describes the use of aerosol optical depth (AOD) and carbon monoxide (CO) retrievals from satellite observations to explain the effect of the aging process of biomass burning (BB) aerosol emissions on the enhancement of aerosol mass concentrations in smoke as it is transported downwind, with the aim of improving the representation of these BB aerosol emissions in chemistry transport models (CTMs) and climate models. "The goal of this study is to investigate the feasibility of deriving the information on BB aerosol aging from satellite measurements of AOD and CO columns."

The study found that smoke aging produces enhancement of the smoke aerosol loading (expressed in terms of column aerosol optical depth), which almost doubles within temporal scales of $\sim$10 hours, especially in dense smoke plume conditions (with PM2.5 concentrations exceeding 100 $\mu$g m-3). However, the enhancement was found to decrease thereafter, although with significant uncertainty, and the study was not able to resolve what happens within the first 5 hours of the BB aerosol emissions. Nevertheless, this study has provided some insight into the evolution of aerosol loading due to aging processes at timescales (> 5 hours) that have not been adequately explored hitherto, thereby contributing toward finding possible pathways for resolving one of the outstanding significant uncertainties in model simulations of BB aerosols. The authors have demonstrated thoroughness in conducting sensitivity studies to account for possible uncertainties due to their methodology and assumptions. The manuscript is well written, the methodology and results clearly described, and the illustrations of good quality. Therefore, I believe that this study merits publication and is appropriate for Atmospheric Chemistry and Physics (ACP).

However, the authors should address some (mostly minor) issues highlighted in my specific comments.

Specific Comments

In the following comments, I have highlighted a few specific issues that need to be addressed, but certainly not all of them. I suggest that the authors use the identified issues (including typos and grammatical errors) as only examples of things to look out for, as they very carefully read the manuscript to find and correct similar occurrences of such issues or others wherever they exist in the manuscript.

The authors state (Page 19, Lines 1-3) that: "the analysis presented in Fig. 5 clearly indicates that the VBS scheme enables more adequate representation of BB aerosol dynamics than the standard scheme at the first (growing) stage of BB aerosol aging." However, only STN simulations are shown in Figure 4. It would be good to include

(later in the manuscript) a figure showing the spatial visualization (similar to Figure 4) of simulations comparing the results of incorporating the aging process in the model against those that do not consider aging. Such visuals would more readily demonstrate the benefit of this work.

At various points in the article, the authors raise an important issue that needs to be investigated, but immediately state that it is "beyond the scope of this study" (e.g. Page 8 - Line 6, Page 15 - Line 13, Page 18 - Line 5, Page 18 - Line 29). Given that the scope of a study is not set in stone anywhere, but typically determined by the authors themselves, it is unnecessary to identify an essential aspect of an investigation and turn around to say that it is beyond the cope of your study. There is no rule preventing the authors from conducting such analyses in this study. Therefore, I suggest that the authors find a better way to express why they cannot conduct such relevant analyses, make a suggestion on how to effectively approach each of such issues, or avoid raising them in the first place.

Page 4, Line 6: change "doubled" to "increased". You have "by a factor of 2" later in the sentence, which makes the use of "doubled" repetitive.

Page 5, Line 21: delete "and" from "algorithm and is".

Page 6, Line 21-22: delete one "type" from "a given type of land cover type".

Page 8, Line 22: It is not clear what is meant by: "as it is follows from ours simulations". Please rephrase and clarify.

Page 11, Lines 1-2: Unconventional sentence construct: "Only those grid cells and days were considered to be representative of background conditions, where the contribution of the fires to the simulated values of both CO columns and AOD did not exceed 10 percent." Please rephrase.

Page 12, Line 9: insert "to" after "corresponding".

Page 15, Line 5: delete "of" before "parameters".

Page 17, Line 17: replace "adequate" with "reasonable". Since the differences between measured and modeled values are still apparently significant, these results should not be described by the term "adequate".

Page 21, Line 31: There are no "green crosses" in Figure 7b. The crosses are black.

Page 24, Line 2: It is not clear how "absorption" can increase the surface area of aerosol particles. Please explain the physical mechanism implied here. I think you probably mean "hygroscopicity" (which involves the absorption of moisture that may cause aerosol particle to swell). However, "absorption" is not the technical term used to describe that process. "Absorption" is mostly used to refer to light absorption (as opposed to "scattering").

Page 26, Line 23: change "then unity" to "than unity".

---

## Referee Comment (RC2) · Anonymous Referee #1 · 7 Dec 2016

The manuscript explores the effect of aging on the evolution of aerosol optical depth (AOD) of biomass burning smoke. The authors use a combination of satellite observations of AOD and CO column amount (with MODIS and IASI respectively) with CHIMERE model simulations of the same variables, and calculate AOD enhancement ratio (ER). Positive ER values mean that BB aerosol that has undergone the aging process is optically thicker than when originally emitted. The implication of this finding is that when aerosol models do not consider secondary aerosol processes, it results in model underestimation of total column AOD in the BB-affected locations. Introduction of aerosol aging process into CTMs could improve simulations of BB aerosol properties.

[Figure]

The submitted manuscript addresses an important topic within the active area of research. The reasons for discrepancy between model and satellite BB AOD are currently an area of vivid interest in the modeling and satellite observation community. Investigation of aging process as one of the factors explaining observed discrepancies is very timely. I thank the authors for a comprehensive, well-structured and thorough work. The paper is of adequate quality to be published in ACP with revisions and clarifications outlined below.

Comments on the content:

P4 lines 11-14: The sentence starting with "Konovlaov et al. (2014) optimized BB emissions..." and till the end of the paragraph is very difficult to read. I had trouble understanding if the values of the factors 2.2 and 2.9 refer to emissions or to ratios of optimal BB emissions. Please rephrase.

P9 lines 10-24: I understood from this section (specifically P9 lines16-17: "...we zeroed emissions of other types of aerosol and disabled secondary aerosol formation from anthropogenic and biogenic precursors.") that model runs including BB emissions (VBS and STN) only include BB aerosols without the other background aerosol. Does this mean that to obtain total AOD for, say, VBS run you add VBS_AOD and BGR_AOD? If so, then I can't see where this is stated explicitly (or this is what P10 lines 30-34 mean?). Please clarify.

P11 lines 14-15: Should T1 read T0? From the definition of trace species on P8 line 20, it looks like T0 is the only chemically passive species, the concentration of which will stay constant over time in the presence of OH. If I am mistaken and you insist that there should be T1 please elaborate (on P11 line 14, after the words "would be constant") on why this is so.

P12 line 23: what is BB part of the observed AOD? Is this total MODIS AOD less the background AOD? In which case, is this observed background AOD value obtained by averaging the most background-like values as described in P10 line 27 – P 11 line 2?

[Figure]

Also, from here on in the text, only one kind of background values are referenced. These are all the variables with subscript b. It is not very clear to me if these background values are all from the model, or there are separate backgrounds for model and observations (e.g. in Eq. 9 are the same values subtracted from the first (obs) and the second (model) parts of the terms in the parentheses?), or it is some combined value applicable to both model and observation? I see that this is explained somewhat afterwards (P13 line 25 till the end of the section), but this still does not answer the question of using the same or different backgrounds for model and observation in the same equation.

Suggested technical changes:

P5 line 9: "ageing" should be "aging". I noticed this instance, but please check the rest of the manuscript for consistency.

P5 line 21: remove "and" between "algorithm" and "is".

P8 line 23: remove either 'is' or 'it is' between the words 'as' and 'follows'.

P16 line 9: remove "and" between "comparison" and "of the measurement".

Fig. 2b: Y-axis label contains subscript bm, which is not used anywhere else in the paper. Please make consistent or explain.

Fig.5 in the headers of the figures: dt is nowhere defined. From the text in section 3.2, I deducted that this is probably the bin width, but could be helpful of the notation was also mentioned.

---

## Author Comment (AC1) · 30 Jan 2017

We are very grateful to the Referee for the positive evaluation of our manuscript and for the useful comments which are carefully addressed in the revised manuscript. Below we describe our point-to-point responses to the referee's comments.

*Referee's comment*: *P4 lines 11-14: The sentence starting with "Konovalov et al. (2014) optimized BB emissions..." and till the end of the paragraph is very difficult to read. I had trouble understanding if the values of the factors 2.2 and 2.9 refer to emissions or to ratios of optimal BB emissions. Please rephrase.*

The values of the factors 2.2 and 2.9 refer to ratios of the optimal BB emissions. We

understand that the sentence indicated by the referee was difficult to read. It has been rephrased in the revised manuscript.

*Referee's comment: P9 lines 10-24: I understood from this section (specifically P9 lines16-17: ". . . we zeroed emissions of other types of aerosol and disabled secondary aerosol formation from anthropogenic and biogenic precursors.") that model runs including BB emissions (VBS and STN) only include BB aerosols without the other background aerosol. Does this mean that to obtain total AOD for, say, VBS run you add VBS_AOD and BGR_AOD? If so, then I can't see where this is stated explicitly (or this is what P10 lines 30-34 mean?). Please clarify.*

Indeed, to obtain total AOD, we added AOD from the BGR run and AOD from the STN (or VBS) run. It was not explained explicitly, and we are sorry for this omission. The corresponding explanation is introduced at the end of the first paragraph in Sect. 2.2.2.

*Referee's comment: P11 lines 14-15: Should $T_1$ read $T_0$? From the definition of trace species on P8 line 20, it looks like $T_0$ is the only chemically passive species, the concentration of which will stay constant over time in the presence of OH. If I am mistaken and you insist that there should be $T_1$ please elaborate (on P11 line 14, after the words "would be constant") on why this is so.*

Yes, $T_1$ should read $T_0$, while $T_2$ should read $T_i$. We are sorry for the confusion. The corresponding typos are corrected in the revised manuscript.

*Referee's comment: P12 line 23: what is BB part of the observed AOD? Is this total MODIS AOD less the background AOD? In which case, is this observed background AOD value obtained by averaging the most background-like values as described in P10 line 27 - P11 line 2?*

Indeed, the BB part of the observed AOD is, by definition, the total MODIS AOD less the background AOD. And yes, the background AOD value can be estimated by averaging the most background-like values selected in the same way as described in P10 line

27 - P 11 line 2 in the reviewed manuscript. To avoid possible misunderstanding, we have slightly modified the corresponding text. Specifically, we removed the words "the BB part" and indicated that the background part of AOD was evaluated by using both the measurement and simulation AOD data as explained in the same section below (specifically, the corresponding explanation is provided in the second paragraph after Eq. 9).

*Referee's comment: Also, from here on in the text, only one kind of background values are referenced. These are all the variables with subscript b. It is not very clear to me if these background values are all from the model, or there are separate backgrounds for model and observations (e.g. in Eq. 9 are the same values subtracted from the first (obs) and the second (model) parts of the terms in the parentheses?), or it is some combined value applicable to both model and observation? I see that this is explained somewhat afterwards (P13 line 25 till the end of the section), but this still does not answer the question of using the same or different backgrounds for model and observation in the same equation.*

In our analysis, the background values are not available directly from observations but can be evaluated only by using a model for selection of the grid cells and days corresponding to the background-like conditions. On the other hand, the modeled background values can be affected by biases that are corrected by using the observational data as explained in Sect. 2.2.2. Thus, it was not feasible to evaluate the background values for the modeled and observed AOD separately. Rather, we used the combined values (obtained in the two different ways as explained in the second paragraph after Eq. 9) applicable to both measurements and observations. In the revised manuscript, we clarify (by means of a statement following after Eq. 9) that the background values, $\nu_b$ and $\tau_b$, of the CO columns and AOD are the same for the corresponding observations and simulations Eq. (9).

*Referee's comments: Suggested technical changes:*

*P5 line 9: "ageing" should be "aging". I noticed this instance, but please check the rest of the manuscript for consistency.*

*P5 line 21: remove "and" between "algorithm" and "is".*

*P8 line 23: remove either 'is' or 'it is' between the words 'as' and 'follows'.*

*P16 line 9: remove "and" between "comparison" and "of the measurement".*

*Fig. 2b: Y-axis label contains subscript bm, which is not used anywhere else in the paper. Please make consistent or explain.*

*Fig.5 in the headers of the figures: dt is nowhere defined. From the text in section 3.2, I deducted that this is probably the bin width, but could be helpful of the notation was also mentioned.*

We thank the referee for the suggested technical changes, all of which are introduced in the revised manuscript. Note that owing to the referee's comment, we noticed some inconsistency in the notations. Specifically, while the width of the photochemical age bin was originally denoted as $d_a$, in some instances (including the headers of several figures), the same value was mistakenly referred to as $d_t$. Both the text and figures are corrected accordingly in the revised manuscript.
* * *

---

## Author Comment (AC2) · 30 Jan 2017

We thank the Referee for the positive evaluation of our manuscript and for the helpful suggestions. All of them are carefully addressed in the revised manuscript. Below we describe our point-to-point responses to the referee's comments.

*Referee's comment: . . . I have highlighted a few specific issues that need to be addressed, but certainly not all of them. I suggest that the authors use the identified issues (including typos and grammatical errors) as only examples of things to look out for, as they very carefully read the manuscript to find and correct similar occurrences of such issues or others wherever they exist in the manuscript.*

[Figure]

We are sorry for any grammatical errors and typos that we did not notice before submission of the reviewed manuscript. We have followed the referee's suggestion and carefully re-read the manuscript. Corresponding corrections are made in the revised manuscript.

*Referee's comment: The authors state (Page 19, Lines 1-3) that: "the analysis presented in Fig. 5 clearly indicates that the VBS scheme enables more adequate representation of BB aerosol dynamics than the standard scheme at the first (growing) stage of BB aerosol aging." However, only STN simulations are shown in Figure 4. It would be good to include (later in the manuscript) a figure showing the spatial visualization (similar to Figure 4) of simulations comparing the results of incorporating the aging process in the model against those that do not consider aging. Such visuals would more readily demonstrate the benefit of this work.*

Our decision not to include a figure showing the AOD spatial distribution according to the VBS simulation in the reviewed manuscript was made by taking into account that it was similar to that depicted in Sect. 4d (as noted in Sect. 3.1). However, the referee's comment indicates that the omission of this figure was not sufficiently justifiable. Accordingly, the missing distribution has been included in the revised manuscript. To do it in the optimal way, we have split the original Fig. 4 into two figures, one of which (Fig. 4) shows only CO columns, while another demonstrates AOD distributions. Furthermore, we have provided additional plots showing the results of the VBS simulation for the two selected days (21 and 22 July, 2012) in the Supplementary material (see Fig. S3). The extension of Fig. S3 allowed us to visualize some improvement in the agreement between the spatial distributions of the measured and simulated AOD values due to the use of the VBS scheme instead of the standard one.

The fact that a direct comparison of the AOD simulations and observations (when they are averaged temporally or spatially) did not allow us to tell which of the model configurations is more adequate (as it is explained in Sect. 3.1) emphasizes the benefits of the method introduced in our paper. Indeed, in contrast to a conventional comparative analysis, our statistical consideration of the AOD enhancement ratio as a function of the BB aerosol photochemical age demonstrated quite clearly that the VBS scheme enables more adequate representation of BB aerosol dynamics than the standard scheme at the first (growing) stage of BB aerosol aging. This result allows us to believe that our method provides a convenient tool to visualize the effects of BB aerosol aging in the simulated data (as noted at the end of Sect. 4).

*Referee's comment: At various points in the article, the authors raise an important issue that needs to be investigated, but immediately state that it is "beyond the scope of this study" (e.g. Page 8 - Line 6, Page 15 - Line 13, Page 18 - Line 5, Page 18 - Line 29). Given that the scope of a study is not set in stone anywhere, but typically determined by the authors themselves, it is unnecessary to identify an essential aspect of an investigation and turn around to say that it is beyond the scope of your study. There is no rule preventing the authors from conducting such analyses in this study. Therefore, I suggest that the authors find a better way to express why they cannot conduct such relevant analyses, make a suggestion on how to effectively approach each of such issues, or avoid raising them in the first place.*

Indeed, we mentioned several points which, in our opinion, deserve careful consideration in the framework of dedicated studies, and we agree that it was unnecessary. Accordingly, following the reviewer's suggestion, we have tried to avoid raising such points. We believe that the corresponding stylistic changes did not affect the overall quality of the scientific discussion.

*Referee's comment: Page 4, Line 6: change "doubled" to "increased". You have "by a factor of 2" later in the sentence, which makes the use of "doubled" repetitive.*

The sentence is corrected in the revised manuscript as suggested by the referee.

*Referee's comment: Page 5, Line 21: delete "and" from "algorithm and is".*

The misprint is corrected in the revised manuscript.

*Referee's comments: Page 6, Line 21-22: delete one "type" from "a given type of land cover type".*

The corresponding sentence is re-phrased to avoid the repetitive use of the words "land cover types".

*Referee's comments: Page 8, Line 22: It is not clear what is meant by: "as it is follows from ours simulations".*

We meant that the indicated mean value of the OH concentration in BB plumes was obtained from our model results. In the revised manuscript, instead of referring to our simulations (which are described only after the sentence in question), we provided a reference to relevant experimental results.

*Referee's comments: Page 11, Lines 1-2: Unconventional sentence construct: "Only those grid cells and days were considered to be representative of background conditions, where the contribution of the fires to the simulated values of both CO columns and AOD did not exceed 10 percent.*

The sentence criticized by the referee has been rephrased as follows: "A given grid cell and day was assumed to be representative of background conditions only if the BB fractions in the simulated values (based on the STN and BGR model runs) of both CO columns and AOD did not exceed 10 percent."

*Referee's comments: Page 12, Line 9: insert "to" after "corresponding". Page 15, Line 5: delete "of" before "parameters".*

The suggested correction is inserted in the revised manuscript.

*Referee's comments: Page 17, Line 17: replace "adequate" with "reasonable". Since the differences between measured and modeled values are still apparently significant, these results should not be described by the term "adequate".*

The suggested change is made in the revised manuscript.

*Referee's comments: Page 21, Line 31: There are no "green crosses" in Figure 7b. The crosses are black.*

The mistake is corrected in the revised manuscript: the word "green" is replaced with the word "black".

*Referee's comments: Page 24, Line 2: It is not clear how "absorption" can increase the surface area of aerosol particles. Please explain the physical mechanism implied here. I think you probably mean "hygroscopicity" (which involves the absorption of moisture that may cause aerosol particle to swell). However, "absorption" is not the technical term used to describe that process. "Absorption" is mostly used to refer to light absorption (as opposed to "scattering").*

We meant that liquid or amorphous aerosol particles can grow as a result of absorption (uptake) of both organic and inorganic compounds from the gas phase (according to the partitioning equilibrium theory). However, we agree that the term "absorption" is mostly used to refer to optical properties of aerosol. Accordingly, in the revised manuscript, we replaced it with a more general (and more conventional) term "condensation".

*Referee's comments: Page 26, Line 23: change "then unity" to "than unity".*

The suggested correction is made in the revised manuscript.